# Trace is the Next AutoDiff: Generative Optimization with Rich Feedback, Execution Traces, and LLMs

**Ching-An Cheng**[*]
Microsoft Research
chinganc@microsoft.com

**Allen Nie**[*]
Stanford
anie@cs.stanford.edu

**Adith Swaminathan**[*]
Netflix
aswaminathan@netflix.com

## Abstract

We study a class of optimization problems motivated by automating the design and update of AI systems like coding assistants, robots, and copilots. AutoDiff frameworks, like PyTorch, enable efficient end-to-end optimization of differentiable systems. However, general computational workflows can be non-differentiable and involve rich feedback (e.g. console output or user's responses), heterogeneous parameters (e.g. prompts, codes), and intricate objectives (beyond maximizing a score). We investigate *end-to-end generative optimization* – using generative models such as LLMs within the optimizer for automatic updating of general computational workflows. We discover that workflow execution traces are akin to back-propagated gradients in AutoDiff and can provide key information to interpret feedback for efficient optimization. Formally, we frame a new mathematical setup, Optimization with Trace Oracle (OPTO). In OPTO, an optimizer receives an execution trace along with feedback on the computed output and updates parameters iteratively. We provide a Python library, Trace, that efficiently converts a workflow optimization problem into an OPTO instance using PyTorch-like syntax. Using Trace, we develop a general LLM-based generative optimizer called OptoPrime. In empirical studies, we find that OptoPrime is capable of first-order numerical optimization, prompt optimization, hyper-parameter tuning, robot controller design, code debugging, etc., and is often competitive with specialized optimizers for each domain. We envision Trace as an open research platform for devising novel generative optimizers and developing the next generation of interactive learning agents. Website: https://microsoft.github.io/Trace/.

## 1 Introduction

Computational workflows that integrate large language models (LLMs), machine learning (ML) models, orchestration, retrievers, tools, etc., power many state-of-the-art AI applications [1]: from chatbots [2], coding assistants [3], robots [4], to multi-agent systems [5]. However designing a computational workflow requires laborious engineering because many heterogeneous parameters (e.g. prompts, orchestration code, and ML hyper-parameters) are involved. Moreover, after deployment any erroneous behaviors of the workflow persist unless a developer manually updates it.

We study a class of optimization problems motivated by automating the design and update of computational workflows. Computational workflows produce optimization problems with heterogeneous parameters, rich feedback (e.g. console output and user's verbal responses), and intricate objectives (beyond maximizing a score). Moreover, a workflow can have interdependent steps (e.g. adaptive orchestration, feedback control loops) and/or involve non-differentiable, semi-black-box, stochastic operations (e.g. ML models, simulations) whose behavior cannot be succinctly captured. As a result, the structure of the computation may change as the parameters and the inputs of the workflow vary.

---

[*]Equal contribution

38th Conference on Neural Information Processing Systems (NeurIPS 2024).

Due to its complexity, computational workflow optimization is usually framed as a black-box [6] or algorithm configuration [7] problem, and is tackled by general techniques like Bayesian Optimization [8], Evolutionary Algorithms [9], Reinforcement Learning (RL) [10] using scalar scores as feedback. But one observation of scalar feedback alone does not provide an improvement signal, so these algorithms are very inefficient when the parameter space is large (e.g. codes or natural language prompts). Recently LLM-based optimizers [11–16] have been proposed as *generative optimizers* to improve efficiency, leveraging the prior of generative models learned from large pre-training corpora to optimize complex prompts and codes. In this paper, we investigate how *generative optimization* can be applied more broadly and systematically to optimize a general computational workflow end-to-end. Appendix B discusses related works in generative optimization.

## 1.1 Toward Efficient End-to-End Optimization of Computational Workflows

Inspired by back-propagation [17], we take an end-to-end approach to computational workflow optimization. AutoDiff frameworks [18, 19] have scaled the back-propagation algorithm to optimize differentiable workflows (e.g., neural networks) with billions of parameters. We extend the idea of AutoDiff and design a new framework, Trace, for jointly optimizing all of the heterogeneous parameters in general computational workflows, which may not be differentiable.

Trace treats a computational workflow as a graph like a neural network, where nodes are either inputs, parameters or the results of computation steps, and directional edges denote how nodes are created from others. But, instead of gradients, Trace propagates the *execution trace* of a workflow (which records the intermediate computed results and how they are used), via the notion of minimal subgraph (see Section 3.3). We show that propagating the execution trace subsumes back-propagation for differentiable workflows, and remains applicable even for non-differentiable workflows. Viewing a workflow as a graph and using its execution trace is standard for software engineering; for instance, human developers use such traces to debug distributed systems [20]. Our novel insight is that execution traces also enable end-to-end generative optimization, because they provide the information needed to relate rich feedback to the parameters in general workflows.

## 1.2 Example of Trace in Action

Trace is available as a Python library with an API inspired by PyTorch [19]. A user declares the parameters to be optimized using a `trainable` flag, decorates the workflow with `node` and `bundle` wrappers, and runs a Trace optimizer – just like how they would declare and train neural networks.

Consider building an AI agent for the Battleship game (Fig. 1). The agent's policy (Fig. 2a) has two components (`reason` and `act`), which are chained together to react to different board configurations. The Battleship environment provides feedback (binary reward in texts) to tell if the agent's action hit the hidden ships, and the goal is to hit all hidden ships as fast as possible. Consider how a human programmer might approach the problem. They may run the policy and change the code based on the

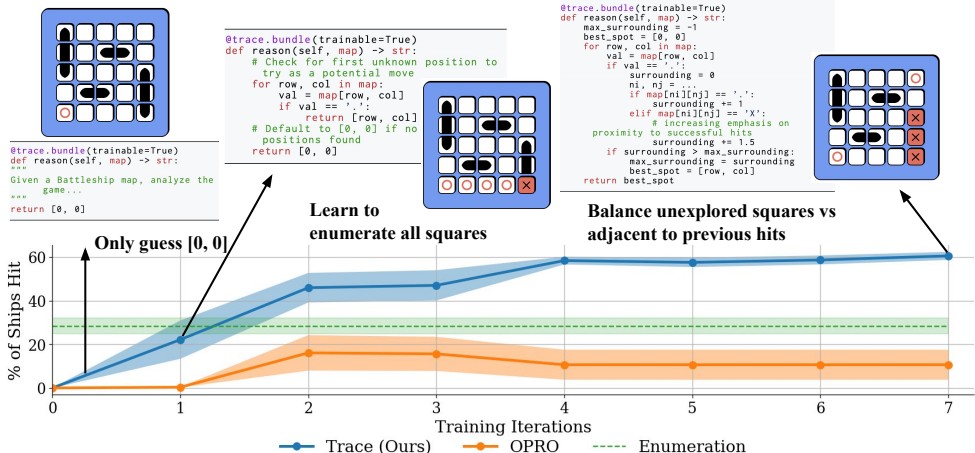

**Figure 1: Learning Example in Battleship.** Trace automatically optimizes heterogeneous parameters (multiple codes) to train an agent to Battleship. Means and standard errors are computed over 10 random seeds.

```
class Policy(trace.Module):

  def forward(self, map):
    plan = self.reason(map)
    output = self.act(map, plan)
    return output

  @trace.bundle(trainable=True)
  def reason(self, map) -> str:
    """
    Given a Battleship map, analyze
      the game...
    """
    return [0, 0]

  @trace.bundle(trainable=True)
  def act(self, map, plan):
    """
    Given a map and plan, select a
      target coordinate...
    """
    return
```

```
policy = Policy()
params = policy.parameters()
optimizer = trace.Optimizer(params)

env = gym.make('Battleship-v0')
board = env.reset()
done, feedback = False, None
while not done:
  # Forward pass
  try:
    target = policy(board)
    board, feedback, done =
      env.step(target.data)
  except TraceExecutionError as e:
    feedback = str(e)
    target = e.exception_node
  # Backward pass and update
  optimizer.zero_feedback()
  optimizer.backward(target, feedback)
  optimizer.step()
```

**(a)** We write a trainable policy in Python using Trace operators.

**(b)** We then use PyTorch-like optimization syntax to train the policy.

**(c)** Trace automatically records execution DAG.

**Figure 2: Python Code of the Battleship Example.** To build a self-adapting agent with Trace, we only need to annotate some empty functions (reason, act) and set up an optimizer following PyTorch semantics. For space, we trim the docstrings of the empty functions with "..." and list them in Appendix J. Trace then builds a DAG as the workflow executes and updates the parameters (see Fig. 1 for the result).

observed feedback. They may rewrite the code a few times to try different heuristics to solve this problem. They will fix any execution errors (e.g., out-of-bounds exceptions) by using stack traces.

Our Trace framework accomplishes the programmer's goal automatically without adding complexity to the Python code. The user declares reason and act as trainable (Fig. 2a) and then runs the agent in a PyTorch-like training loop (Fig. 2b). During the execution, Trace records a directed acyclic graph (DAG) (Fig. 2c) and uses it to compute the execution trace for optimization. Trace also automatically catches errors (e.g., syntax/semantic errors) and can use them as feedback. In Fig. 1, we show what the agent learns as Trace optimizes[2] its policy, where the learned policy is evaluated on new randomly generated games for generalization. The agent can quickly improve its performance and learn strategies that are increasingly complex. Remarkably, there is no mention of the specific Battleship environment API, nor details on how the functions reason and act should behave or adapt in Fig. 2a. The Trace optimizer figures out all the details through interactions as the computational graph unfolds and the feedback on the output is observed. Beyond code as parameters in this example, we also have experiments in Section 5 where prompts and other heterogenous parameters are optimized.

## 1.3 A New World of Optimization

The design of Trace is based on a new mathematical setup of iterative optimization, which we call Optimization with Trace Oracle (OPTO). In OPTO, an optimizer selects parameters and receives a computational graph as well as feedback on the computed output. Trace is a tool to efficiently convert the optimization of computational workflows into OPTO problems in practice.

We argue that framing computational workflow optimization as OPTO can lead to faster convergence than a black-box approach. We present a constructive proof: We design a general-purpose efficient generative optimizer called OptoPrime, for OPTO. OptoPrime turns OPTO to a sequence of pseudo-algorithm problems. In each iteration of OPTO, we format the execution trace and output feedback as a pseudo-algorithm question and present it to an LLM for solution (GPT-4 using a ReAct-CoT prompt listed in Appendix G). In experiments, we apply OptoPrime unchanged to many disparate applications like prompt optimization, first-order numerical optimization, hyper-parameter tuning, and robot controller design. We find that the general purpose OptoPrime is competitive with specialized optimizers for each domain, e.g., achieving $10\%$ higher accuracy on BigBenchHard [21] when optimizing a DSPy [22] program compared to their hand-designed optimizer.

Trace, OPTO, and OptoPrime together provide the first tractable algorithm for optimizing general computational workflows end-to-end. The Trace framework *a)* leverages a workflow's structure and *b)* can incorporate rich feedback beyond scores, extending AutoDiff to complicated, non-differentiable computational workflows. With Trace, we conjecture that "training deep agent networks" (which fluidly mix computation of tensors, LLMs, and other programmable tools) will soon be possible.

---

[2]We use a new general-purpose LLM-based optimizer OptoPrime that we detail in Section 4.

## 2 Optimization with Trace Oracle

OPTO is the foundation of Trace. In this section, we define this graph-based abstraction of iterative optimization and discuss how OPTO covers various computational workflow optimization problems.

**Preliminary** We review the definition of a computational graph (see Fig. 2c). A computational graph $g$ is a DAG, where a node represents an object (such as tensors, strings, etc.) and an edge denotes an input-output relationship. We call a node without parents a root and a node without children a leaf, which are the inputs and outputs of the computational graph. In the context of optimization, some inputs are marked as trainable *parameters*, which are denoted as $\{X_\theta\}$. For a node $X$, its parents are the inputs to an operator that creates $X$. The descendants of node $X$ are those that can be reached from $X$ following the directed edges; the ancestors are defined conversely. Without loss of generality, we suppose that all computational operators have a unitary output[3]. In this way, we can associate the operator that creates the child node with the child node, and the full computation can be represented compactly as a DAG without explicitly representing the operators. The execution trace of a computational workflow is defined as the sequence of operations and their execution results invoked when computing the output from a set of inputs; execution traces can be expressed as a computational graph as defined above.

### 2.1 Problem Definition of OPTO

OPTO is an *abstract* setup of iterative computational workflow optimization. An OPTO problem instance is defined by a tuple $(\Theta, \omega, \mathcal{T})$, where $\Theta$ is the parameter space, $\omega$ is the context of the problem, and $\mathcal{T}$ is a Trace Oracle. In each iteration, the optimizer selects a parameter $\theta \in \Theta$, which can be heterogeneous. Then the Trace Oracle $\mathcal{T}$ returns a *trace feedback*, denoted as $\tau = (f, g)$, where $g$ is the execution trace represented as a DAG (the parameter is contained in the root nodes of $g$), and $f$ is the feedback

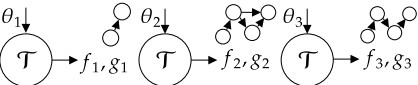

**Figure 3: Iterations of OPTO.** When $\theta \in \Theta$ is selected, the Trace Oracle $\mathcal{T}$ returns trace feedback $\tau = (f, g)$, where $g$ is a computational graph using $\theta$ as input and $f$ is the feedback given to the output of $g$.

provided to exactly one of the output nodes of $g$. Finally, the optimizer uses the trace feedback $\tau$ to update the parameter according to the context $\omega$ and proceeds to the next iteration, as shown in Fig. 3.

In OPTO, the output feedback $f$ is generic, such as scores, gradients, hints/explanations expressed in natural language, and console messages. The context $\omega$ provides invariant information to interpret the output feedback $f$ as well as any known side-information, e.g., desired properties of the parameters. The context $\omega$ is fixed for an OPTO problem instance (similar to an instruction, or a problem definition), whereas the output feedback $f$ can change with the parameter $\theta \in \Theta$ and the resulting computation $g$. For example, $\omega$ may be "Minimize a loss function" and $f$ is a loss. Alternatively, $\omega$ can be open-ended, like "Follow the feedback" and $f$ describes how an output should be changed. In Section 3.2, we discuss how to define the context and output feedback when constructing OPTO problems in practice. In this paper, we focus on OPTO problems where $f$ and $\omega$ can be expressed compactly in text. This covers a wide range of problems [23], including those with scalar feedback.

OPTO differs from a black-box setup in that the execution trace $g$ shows the computational path toward the output, which provides information to construct a parameter update direction from $f$ and $\omega$. In the minimization example above, when the execution trace $g$ is missing, it is unclear how the parameter can be improved given only a point evaluation of $f$. On the other hand, with $g$ documenting the functions used to create the output, an update direction (e.g., a gradient) can be derived. We highlight that the structure of the computational graph $g$ returned by the Trace Oracle $\mathcal{T}$ can be different in each iteration in the general case (as in Fig. 3) because some workflows can change with different inputs and parameters.

To ground the OPTO setup, we show how OPTO is related to some existing problems with examples. We discuss other examples like hyperparameter tuning and multi-agent systems in Appendix C.

**Example 1** (Neural network with back-propagation). The parameters are the weights. $g$ is the neural computational graph and $f$ is the loss. An example context $\omega$ can be "Minimize loss". The back-propagation algorithm can be embedded in the OPTO optimizer, e.g., an OPTO optimizer can use $\tau$ to compute the propagated gradient at each parameter, and apply a gradient descent update.

---

[3]A multi-output operator can always be modeled by a single-output operator and single-output indexers.

**Example 2** (RL). The parameters are the policy. $g$ is the trajectory (of states, actions, rewards) resulting from running the policy in a Markov decision process; i.e., $g$ documents the graphical model of how a generated action is applied to the transition, which then returns the observation and reward. $f$ can be a success flag. $\omega$ can be "Maximize return (cumulative rewards)" or "Maximize success".

**Example 3** (Prompt Optimization of an LLM Agent). The parameters are the prompt of an LLM workflow. $g$ is the computational graph of the agent and $f$ is the feedback about the agent's behavior (which can be scores or natural language). $\omega$ can be "Maximize score" or "Follow the feedback".

## 3 Trace: The Next AutoDiff

We design a framework, Trace, to bring OPTO from an abstract concept to reality. Trace provides a lightweight Python[4] tool to implement the Trace Oracle of OPTO for optimizing computational workflows. Through the OPTO framing, Trace separates the design of optimizers and domain-specific components so that general-purpose optimizers can be built that work across diverse domains.

### 3.1 Design of Trace

Trace is designed based on two primitives:

- node is the wrapper of Python objects. When wrapped, a Python object is registered as a unique node in the computational graph of Trace. A node can be set trainable, which would make the node a parameter in OPTO. In addition, when using node to declare a parameter, one can also describe (in natural language) constraints that the parameter should obey.
- bundle is the decorator to turn Python methods into operators. When a function is decorated, its docstring and source code are recorded as the definition of the operator, which infer how the output feedback should change the parameters. Moreover, functions decorated by bundle can be set trainable as well, which means that the code of the decorated method becomes a parameter.[5]

For any workflow, using Trace involves the following steps (see Fig. 2). First, the user declares the workflow's parameters using node and bundle, and defines the workflow's conceptual blocks as operators using bundle. Then the user creates an OPTO optimizer (such as OptoPrime in Section 4), and optionally provides the context $\omega$ for the problem. (A default context $\omega$ of OptoPrime is "Follow the feedback"). In addition, the user defines a mechanism to provide feedback to the computed result (e.g., scores, natural language suggestions, etc.), in analogy to defining a loss function in neural network training. After the declaration, optimization via Trace repeats the following: *1)* Execute the decorated workflow. As it runs, a DAG is built in the backend, logging the computed results and their connections. *2)* Initiate the propagation of the output feedback to the parameters by calling backward. (Any execution error is also treated as feedback; see Appendix D.) Internally, Trace extracts the minimal subgraph $g$ connecting the parameters and the output and sends the OPTO optimizer the trace feedback $\tau = (f, g)$. *3)* Call the OPTO optimizer's step method to update the parameters.

### 3.2 Using Trace Primitives for Effective Execution Tracing

There are many ways to represent a computational workflow as a computational graph, from abstracting the entire process as one big operator to listing all low-level steps as operators in the graph. In Trace, the level of abstraction is decided by how bundle is applied, as all steps underneath bundle are abstracted as one operator. The design of bundle allows tracing most Python codes, except for those modifying the content of an object reference in place. However, such a case can be avoided by first duplicating the object and then applying the modification to the copied object, similar to how a recurrent neural network is implemented.

Different abstraction choices trade off the graph complexity and the description needed for each operator. Abstracting everything into a single operator makes a trivial graph but requires more descriptions to faithfully capture the workflow. On the other hand, not all details matter in optimization, so exposing every low-level operator can make the graph unnecessarily cluttered. The best representation depends on the application and OPTO optimizer at hand. This problem, we believe, is similar to the design of neural network architectures. Here, we suggest defining the operators to

---

[4]The current implementation is in Python. The DAG-based design of Trace can be adapted to other programming languages also, just like AutoDiff libraries that are available in C++.

[5]This would add an extra parent (i.e., the trainable code) to the computed child node.

mimic the whiteboard system diagram of the computational workflow. This level of abstraction in our experiments strikes a good balance between the ease of documentation and the graph size.

Apart from architecture design, another design question is what information goes into the context $\omega$ versus the description of each operator? For a *single* problem, there is no difference in principle; one can choose to provide details of all operators in $g$ through the context $\omega$. However, this will require manually crafting a context for every workflow. We suggest instead providing a description of the operators when they are defined using bundle. Then Trace will automatically generate the workflow-specific information while the same context $\omega$ is shared across *many* workflows.

### 3.3 Backward Feedback Propagation: Realizing the Trace Oracle of OPTO

Trace uses a recursive graph traversal algorithm (Algorithm 1) to propagate feedback in the reversed topological ordering. With different propagators, Algorithm 1 can implement various forward-backward schemes including back-propagation.[6] We propose a general propagator, Minimal Subgraph Propagator (MSP), in Algorithm 2. MSP propagates the trace feedback $\tau = (f, g)$, where $g$ is implemented as a priority queue. Running Algorithm 1 with MSP (Algorithm 2) together implements the Trace Oracle of OPTO, which extracts the *minimal subgraph* between parameter nodes and output.[7] connecting the parameters and an output. Appendix E proves the following:

**Theorem 1.** *For a graph with $N$ nodes and maximum degree $W$, Algorithms 1 and 2 have time complexity $O(WN^2 \log N)$ and space complexity[8] $O(WN)$.*

By contrast, back-propagation has time and space complexities of $O(WNd^2)$ and $O(d)$ respectively, where $d$ is the maximal dimension of tensors. The difference is because in the most general setting of computational graphs and feedback, the propagated feedback (no matter how it is represented) does not have a constant size and needs the full subgraph.

---

**Algorithm 1** Backward Message Passing

**Input:** Node $output$, feedback $f$, propagator $P$
1: $\tau \leftarrow P.\text{init}(f)$
2: $output.\text{add\_feedback}(\text{"User"}, \tau)$
3: $queue \leftarrow \text{MinHeap}([output])$
4: **while** $queue$ is not empty **do**
5: $\quad node \leftarrow queue.\text{pop}()$
6: $\quad feedback \leftarrow P.\text{propagate}(node)$
7: $\quad$ **for** $parent$ **in** $node.\text{parents}$ **do**
8: $\quad\quad \tau \leftarrow feedback[parent]$
9: $\quad\quad parent.\text{add\_feedback}(node, \tau)$
10: $\quad\quad$ **if** $parent \notin queue$ **then**
11: $\quad\quad\quad queue.\text{push}(parent)$

---

**Algorithm 2** Minimal Subgraph Propagator

**Input:** A child node $node$
$\quad$ // The pseudo code implements propagate.
$\quad$ // init(f) returns $(f, \{\})$.
1: $g \leftarrow \{node\} \bigcup \{parent \text{ in } node.\text{parents}\}$
2: **for** $(f_i, g_i)$ in $node.\text{feedback}$ **do**
3: $\quad g \leftarrow g \bigcup g_i$
4: $\quad f \leftarrow f_i \quad$ // all $f_i$ are the same.
5: **return** $\{p : (f, g) \text{ for } p \text{ in } node.\text{parents}\}$

---

**Theorem 2.** *For generic computational graphs of $N$ nodes, in the worst case, the propagated feedback needs a description length $\Omega(N)$ to construct an improvement direction.*

Despite the worst case complexity of MSP, in practice the difference is negligible. Since MSP only involves merging priority queues of references, most actual computation happens in the forward pass (and also the optimizer's step method). However for very large problems with millions of nodes in the minimal subgraph, we anticipate that computational issues of MSP could arise.

## 4 Design of the First OPTO Optimizer

We introduce an LLM-based generative optimization algorithm OptoPrime for any text-based OPTO problem. We believe that this is one of many possible optimization algorithms for these problems and there is a large space to be explored for identifying efficient optimization methods for OPTO.

**Subgraph Representation** One core challenge of designing an LLM-based OPTO optimizer is how to represent the execution trace subgraph $g$ (which can involve various graph structures and heterogenous data) to LLMs, in a way that LLMs can understand and reason about the downstream effects of parameter updates. We leverage the LLMs' remarkable coding and debugging ability [3].

---

[6]In back-propagation, the message is the gradient $\nabla_i$ and the propagate function returns $J_i^\top \sum_j \nabla_j$ to its $i$th parent, where $J_i$ is the Jacobian to the $i$th parent and and the $\nabla_j$ gradient received from the $j$th child.

[7]The *minimal subgraph* $g_{\mathcal{X} \to Y}$ between nodes $\mathcal{X}$ and a node $Y$ is defined as $g_{\mathcal{X}, Y} := \mathcal{X} \bigcup \{Y\} \bigcup \{Z, P | Z \in \text{ancestors}(Y), Z \in \text{descendants}(X), X \in \mathcal{X}, P \in \text{parents}(Z), P \in \text{parents}(Y)\}$.

[8]The space complexity refers to the extra space needed for the backward pass, not including the forward pass.

We present the trace feedback $(f, g)$ computed by Trace as a pseudo-algorithm problem: the subgraph $g$ is expressed as a report of code execution with information about the computed values and descriptions of operators in $g$. Then we prompt the LLM to update the parameters in $g$ based on feedback $f$ given to the output. Fig. 4 shows an example. It is crucial to note that even though the lines look like an actual program, it is not the real program but the computational graph defined by bundle (see Section 3.2).

**Parameter Update**   We prompt the LLM with a ReAct-CoT style prompt (Appendix G.2) in one query, asking it to generate reasoning based on the graph, and a suggestion on the parameter changes. If the suggestion can be extracted from the response, we update the parameters.

**Optimization Memory**   OptoPrime optimizes most workflows reasonably well using just the traced graph and feedback, but it can run into issues when single feedback alone is not informative (e.g., the output feedback is rewards, but there is no description of how the rewards are generated). For robustness, we have a basic memory module in OptoPrime, which tracks the past parameter-feedback pairs as in-context examples. See Appendix G for details.

```
#Code:
a = bar(x)
y = add(b, a)
z = mul(a, y)
#Definitions:
[mul] This is a multiply operator.
[add] This is an add operator.
[bar] This is a method that does
    negative scaling.
#Inputs:
b=1.0
#Others:
a=2.0
y=3.0
#Output
z=6.0
#Variable
x=-1.0
#Feedback:
Output should be larger.
```

**Figure 4:** An example pseudo-code report generated by Trace for a program of x = Node(-1.0); z = bar(x) * (bar(x)+1) and the objective of $\max_x z$.

# 5   Experiments

We evaluate the Trace framework with OptoPrime. We implement the state-of-the-art LLM optimizer OPRO [13] as a baseline; in comparison with OptoPrime, OPRO does not use the execution trace but relies on the memory of parameter and feedback pairs. For these experiments, we use GPT-4-0125-Preview. We run the experiments on a standard PC with 16 GB RAM, and Trace introduces no measurable overhead on executing the workflow. We also conduct experiments to compare Trace and OptoPrime with a concurrent AutoDiff-like framework, TextGrad [24], which was released after Trace was submitted to NeurIPS. We show that TextGrad can be easily implemented as an optimizer in Trace, and OptoPrime achieves similar or better performance than TextGrad while using much less computation time. In the rest of this section, we will denote Trace+OptoPrime simply as Trace. We report the token usages of all approaches in all experiments in Appendix I.1.

## 5.1   Validating with Numerical Optimization

First, we want to validate if OptoPrime can solve classical differentiable optimization problems, since they are a special case of OPTO. Consider the problem of $\min_x |h(x) - y^*|$ for a target $y^*$. We construct a synthetic task environment that randomly creates $y^*$ and the computational graph of $h$ with arbitrarily complex connections between numerical variables (see Appendix I.3 for details). We evaluate OptoPrime (denoted as Trace) and a variant that does not see the graph (Trace Masked); both the optimizers do not use memory. The output feedback is "The output should be <larger/smaller>" (this feedback has the same information as the gradient w.r.t. $h$). We compare also the performance of Adam optimizer [25]. We run 30 trials over different randomly generated problems. All methods see the same randomness. Trace is able to match the best-in-class Adam; on the other hand, without access to the full computational graph, the feedback-alone optimizer struggles to find $x^*$ (Figure 5a).

## 5.2   Tuning Hyperparameters to Orchestrate Complex Systems

We tested Trace in a traffic control problem, which is an instance of hyper-parameter tuning. We used UXSim [26] to simulate traffic at a four-way intersection, where the trainable parameters are 2 integers in [15, 90], which are the green light duration for each direction of traffic flow. The feedback is the estimated delay experienced by all vehicles due to intersections, and the goal of an optimizer is to minimize the delay using the fewest number of traffic simulations. To this end, this optimizer must find the right trade-off for temporally distributed and variable demands. In Fig. 5 we report the performance of a SOTA heuristic from the traffic control literature, SCATS [27] as well as two black-box optimization techniques: Gaussian Process Minimization (GP) [8] and Particle Swarm Optimization (PSO) [28]. All methods use the same starting parameters. Trace denotes OptoPrime using memory, and Trace NoMem denotes OptoPrime without memory. We report further details in

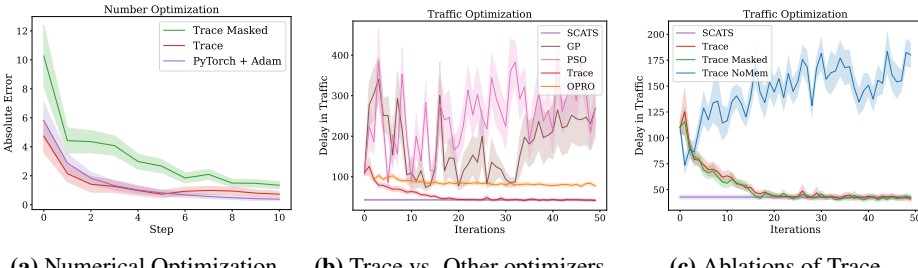

**(a)** Numerical Optimization. **(b)** Trace vs. Other optimizers. **(c)** Ablations of Trace.

**Figure 5:** Numerical Optimization and Traffic Optimization Results.

Appendix I.4. GP and PSO appear bad because 50 iterations are insufficient for their convergence; given enough iterations, both will eventually perform well. Trace is quickly competitive with the SCATS heuristic, whereas OPRO is not. Moreover, we find that memory is crucial for Trace to perform well for this task. But we note that Trace consumes extra overhead compared to other methods, since Trace has to materialize the resulting computation graph and query an LLM with effectively a longer prompt than that of OPRO.

## 5.3 Unifying Prompts and Functions Optimization

Many LLM agents today, e.g., specified by LangChain [29] and DSPy [22], have many components. These libraries provide optimization tools to optimize a small portion of their workflows, predominantly the prompt that goes into an LLM call. However, for building self-adapting agents that can modify their own behavior, only allowing the change to one part of a workflow but not others can be limiting. In this experiment, we test Trace's ability in joint prompt optimization and code generation. Specifically, we optimize a given DSPy-based LLM agent and tune its three components: the meta-prompt `prompt_template`, a function `create_prompt` that modifies the prompt with the current question, and a function `extract_answer` that post-processes the output of an LLM call.

We set up an end-to-end prompt-and-code optimization pipeline. We use an automatic evaluation function to compare the LLM's output with the ground truth, which requires the LLM agent to generate outputs not only with the correct answer but also in the correct format (following the guidelines of [30]). We use the Big-Bench Hard dataset [21] (15 examples for training, 5 for validation, and the rest for testing). We compare Trace with DSPy's COPRO module (which optimizes the meta-prompt). In Table 1, we show that Trace is able to optimize a DSPy program beyond what DSPy's COPRO optimizer can, especially on algorithmic tasks. This result shows how Trace can concretely improve upon existing LLM prompt optimization libraries. We show learned codes in Appendix J.

| 0-shot | BBH all (23 tasks) | NLP (12 tasks) | Algorithmic (11 tasks) | 0-shot | BBH all (23 tasks) | NLP (12 tasks) | Algorithmic (11 tasks) |
|---|---|---|---|---|---|---|---|
| DSPy | 41.6 | 53.8 | 32.6 | DSPy + CoT | 70.4 | 73.7 | 68.0 |
| DSPy-PO | 55.3 | 69.0 | 45.2 | DSPy-PO + CoT | 71.6 | 73.9 | 70.0 |
| Trace | **59.5** | **70.9** | **51.1** | Trace + CoT | **78.6** | **75.8** | **80.6** |

**Table 1:** End-to-end workflow optimization for an LLM benchmark (Big-Bench Hard) in 0-shot setup. `CoT` refers to Chain-of-Thought prompting and `PO` refers to DSPy's own prompt optimizer (COPRO). We use Trace to optimize a DSPy program, starting from the same program and prompt template specified by DSPy.

## 5.4 Long-Horizon Robot Manipulator Control

We test the ability of Trace to optimize long-horizon workflows with complex dependencies and to "back-propagate through time". We use Trace to train the controller code (in Python) for a simulated Sawyer robot manipulator. We use the Meta-World environment from LLF-Bench [23] as the simulator and consider three tasks: Reach, Pick-place, and Push. For each task, LLF-Bench provides a task instruction and meaning of the action space, which we use as the context $\omega$ of the OPTO problem. The observation is a Python dictionary of vectors, indicating the end-effector position, the goal position, the gripper status, etc. The action space is a 4-dimensional vector to control the relative position of the end-effector and the gripper state. In each time step, the LLF-Bench Meta-World simulator returns the observation and natural language feedback to guide the robot. An

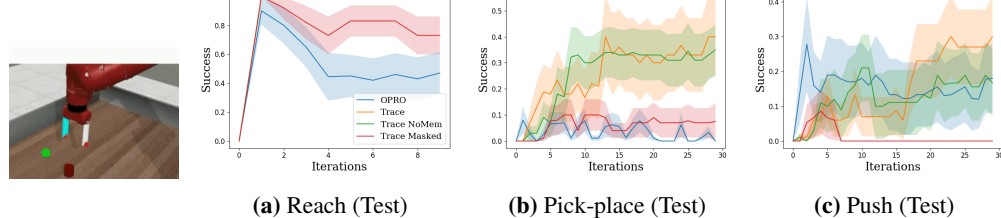

**(a)** Reach (Test)  **(b)** Pick-place (Test)  **(c)** Push (Test)

**Figure 6:** Learning the feedback control policy (code) for a simulated Sawyer manipulator in LLF-Bench Meta-World. In each iteration (x-axis), one episode of rollout (10 steps) is performed, and then the policy is updated. The mean and standard error of the success rate over 10 seeds are shown.

episode ends if the robot successfully solves the problem or because of time-out. We consider an episodic training setting. The initial condition for all iterations in training is the same. We evaluate the learned policy in terms of success, starting from 10 held-out initial conditions. The task horizon is 10 steps, which is sufficient for task completion, and each training iteration has one rollout. The output feedback in OPTO is a textual representation of task success. In addition to the controller code, we also decorated the reset and step functions of the gym environment so that the entire rollout can be traced end-to-end. We compare Trace with OPRO; because of the streaming OPTO setting, our OPRO implementation only proposes one candidate in each iteration, which is then evaluated and provided with the output feedback.

The experimental results are summarized in Fig. 6. Trace denotes OptoPrime using memory, and Trace NoMem denotes OptoPrime without memory. We show learned code in Appendix J. OptoPrime is clearly the top-performing optimizer, especially the version with memory. OPRO is able to solve Reach at the start, but its performance degraded over iterations (this instability was mentioned in [13]) and gets a similar performance as OptoPrime (without memory) in Push. To validate that the performance of OptoPrime is indeed due to using the execution trace, we include an ablation where we mask out the execution trace, which leads to a significant decline in performance and stability. This experiment features the most complex graph structures among all the experiments. The experimental results here are quite impressive, showing that Trace is able to learn a sophisticated control logic in dozens of interactions, not only working on the training initial conditions but also on held-out testing ones too. We discuss some limitations in Appendix I.6.

## 5.5 Comparison with TextGrad

After the submission of our work to NeurIPS, another AutoDiff-like framework, TextGrad [24], was released, which shares the same goal of end-to-end optimizing AI workflows as Trace. In comparison, TextGrad propagates text feedback, whereas Trace propagates minimal subgraphs (see Section 3.3). The graph-based design of Trace, which separates the tracing infrastructure and optimization algorithms, makes it more flexible. In fact, we easily implemented TextGrad as an optimizer in the Trace framework, but the reverse is not possible (because TextGrad couples the infrastructure and the optimization algorithm together). In addition, unlike TextGrad, Trace supports jointly optimizing heterogeneous parameters and can be applied to directly trace a given computational workflow without the need to rewrite the workflow using pre-defined templates. Please see Appendix H for more discussion comparing the two frameworks.

In this experiment, we apply Trace to directly decorate the evaluation code released with the TextGrad library and optimize the parameters following their training/evaluation pipeline line-by-line. This experimental design makes the comparison fair by allowing each optimizer to access the same LLM APIs around the same time, and showcases the flexibility of Trace framework to optimize any computational workflow. We pick the Solution optimization [24, Table 2] and Prompt optimization [24, Table 3] for the reasoning tasks experiments. Please see [24] for details on the exact setup. We compare OptoPrime, TextGrad[9], and a reimplementation[10] of TextGrad as an optimizer in Trace. We find that all these algorithms achieve similar success rates in these experiments. One noticeable difference is that OptoPrime is about 3x faster wall-clock time than TextGrad since OptoPrime makes a single call to LLM in each optimization step, whereas TextGrad calls linear to the graph's size.

---

[9]The numbers in the original paper cannot be reproduced exactly despite using the released TextGrad code.

[10]All algorithms in Trace see the same graph, which is slightly different from the graph of TextGrad.

|  | OptoPrime (Trace) | Time | TextGrad (24-10-30) | Time | TextGrad (Trace) | TextGrad (Reported) |
|---|---|---|---|---|---|---|
| MMLU-Machine Learning | **86.6** (0.2) | 1.7 (0.6) | 86.1 (0.5) | 3.5 (1.1) | 86.3 (0.2) | 88.4 |
| MMLU-College Physics | **94.1** (0.8) | 1.2 (0.3) | 93.1 (0.7) | 2.3 (0.4) | 93.3 (0.6) | 95.1 |
| Google-proof QA | **59.6** (1.3) | 12.2 (1.4) | 53.2 (0.6) | 19.5 (1.9) | 54.0 (0.7) | 55.0 |
| BBH Counting | **89.4** (0.1) | 55.9 (4.5) | 89.2 (1.2) | 142.9 (9.3) | 87.6 (1.7) | 91.9 |
| BBH Word Sorting | 71.6 (3.1) | 82.5 (10.1) | **72.0** (0.4) | 211.1 (16.8) | 71.4 (2.5) | 79.8 |
| GSM8K | **82.5** (0.1) | — | 82.4 (0.6) | — | 82.0 (0.2) | 81.1 |

**Table 2: Comparison between Trace and TextGrad.** The optimizer is GPT-4o-2024-08-06, and the student model is GPT-35-turbo-1106. The results show the mean and the standard error of success rate of the last iterate computed by 5 seeds. The experiment time reported is in minutes (the time involves not just training but also validation and testing by running TextGrad's original pipeline); the time of GSM8K experiment is omitted as the experiment time (>8hrs) is determined primarily by the evaluation not optimization.

## 6 Limitations

We highlight that Trace, OPTO and OptoPrime are a first step towards end-to-end generative optimization and building self-adapting workflows. They have limitations in their current form. OPTO captures rich feedback, but it is important to specify a solution concept as well as the feedback source. We provide guidance for feedback design in Section 3.2 and discuss notions of optimality in Appendix F. We believe designing feedback will be as important as designing loss function in deep learning, both of which are open research questions. Also, Trace cannot convert all computational workflows into OPTO problems, e.g., stateful functions that modify their state in place cannot be represented as a DAG without modification, and distributed/parallel computing workflows are incompatible with the current implementation (though in theory Trace can run in an asynchronous way so long as the overall graph does not end up with cycles). Finally, while Trace is designed to be generic and future-proof, the OptoPrime optimizer is preliminary. Although we demonstrated that OptoPrime could work well with moderate-size graphs, it is not a provably optimal algorithm and uses more tokens than OPRO, though, in our experiments, OPRO's performance does not improve even when given a large token budget. The debugging ability and context limits of the LLM used in OptoPrime crucially determine the scale of problems that we can practically address today. Consequently, more research is needed for designing token-efficient generative optimization algorithms.

## 7 Conclusion and Future Work

We created Trace that can convert a computational workflow optimization problem into an OPTO problem, and we demonstrated a tractable OPTO optimizer, OptoPrime. This is just a first step towards a new paradigm of end-to-end *generative optimization*, with exciting avenues for future work. We discuss a few selected ones below. Please see Appendix A for a longer discussion.

In OptoPrime, we connect optimization to an LLM's reasoning capability. Techniques that have been proposed to improve LLM reasoning, e.g. Chain-of-Thought [31], Few-Shot Prompting [32], Tool Use [33], and Multi-Agent Workflows [5] could also help improve OptoPrime or design new OPTO optimizers. We conjecture that a hybrid workflow of LLM and search algorithms, can enable a truly general-purpose OPTO optimizer. Along the way, we must settle on how to delineate the agent vs. the optimizer. How to trade off the generality of optimizer vs. crafting side-information in the context $\omega$ to achieve task-specific performance is an open question.

In Trace, we chose a specific propagator (MSP), which maximally preserves information for a general computation graph. We can instead specialize it for specific computations, e.g. to accommodate very large graphs with a hierarchical graph representation. Going a step beyond the basic memory module we experimented with in OptoPrime, we anticipate that an optimizer that can reason about how a workflow will behave under counterfactual parameter settings can be more efficient than OptoPrime and can enable a divide-and-conquer approach to OPTO. More research is needed to study the theoretical properties of OPTO (such as optimization landscape and complexity). We hope our preliminary effort in Appendix F can provide some guidance. Finally, in this paper, we focused on output feedback and context that can be compactly textualized. We anticipate that computational workflows with rich non-textual contexts and output feedback will also benefit from automatic generative optimization through appropriate applications of Trace (e.g., with VLMs).

## Acknowledgments and Disclosure of Funding

We would like to thank John Langford, Ahmed Awadallah, Jennifer Neville, Andrey Kolobov, Ricky Loynd and Paul Mineiro for thought-provoking discussions. We would also like to thank Tobias Schnabel, Ruijie Zheng, Wanqiao Xu, and Kaiwen Wang for their valuable feedback on an early draft of this manuscript. Additionally, we thank Anqi Li, Omar Khattab, David Hall, Yifan Mai, Bryan He, Yash Chandak, Emma Brunksill, and Dawen Liang for their suggestions and feedback. The work was partially done while Adith Swaminathan was at Microsoft Research.

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

# A Perspective: Deep Agent Workflows

We posit that the current practice of manually engineering computational workflows to build AI systems is analogous to programmers in the early 2000s hand-coding neural network weights to create engaging AI characters in video games [34]. Just like AutoDiff enabled the automatic and scalable optimization of deep neural networks with billions of parameters, we believe that Trace is the first step towards automatic and scalable optimization of "Deep Agent Workflows" to power even more capable AI systems. However, there are several limitations of the current implementation of Trace that need to be addressed to build Deep Agent Workflows.

When designing interactive AI systems that learn from their interactions, we need to define the *parameters* and *feedback* of the system. Parameters are the internal attributes that can be updated by the learning algorithm employed by the system. Feedback are the things observed and recorded by the system as a product of its interactions, and that provide signal for learning. Trace enables the development of new learning algorithms (e.g. through OptoPrime) that incorporate rich feedback to update heterogenous parameters. In contrast, AutoDiff for deep neural networks uses numerical feedback (e.g. rewards or loss functions) to optimize numerical parameters (e.g. tensors). Black-box optimization techniques (e.g. Reinforcement Learning) can use numerical feedback to optimize heterogenous parameters (e.g. codes, hyper-parameters as well as tensors), though they are inefficient. In the experiments, we saw that Trace was more efficient than black-box methods by using a generalization of back-propagation. Finally, Trace can use rich feedback (e.g. language) to extract more signal for learning.

**What can be traced?** Trace cannot convert all computational workflows into OPTO problems. Workflows with recursive bundle operators or those requiring distributed/parallel computing are not compatible with the current implementation. A future work would be to expand the Trace implementation to support these scenarios. In addition, the current implementation of Trace does not trace the execution within an operator defined by bundle, though in principle this is possible. There are also ambiguities in how an existing workflow can be traced and represented as a DAG. One example is when there is some sub-workflow following an if condition and another one following else. One can choose to wrap the entire code, including if and else, by bundle as a single operator. On the other hand, one can also just wrap the sub-workflows and not trace the if condition nor represent it as part of the DAG. (That is, suppose the if condition is true; from the DAG, one cannot see the alternate path under else). The latter choice has a flavor of applying "stop-gradient" on the boolean condition, whereas the first choice enables back-propagation through also the logical condition. We summarized some design considerations in Section 3.2. We foresee the choice of what to trace and how to trace in building Deep Agent Workflows will be an on-going research problem, similar to neural network architecture design.

**Where do we get rich feedback?** The OPTO framework captures an abstraction of rich feedback, called trace feedback (the execution trace and the output feedback), but it requires specification of operator descriptions and output feedback source to guide the optimization effectively. OPTO can be more efficiently solvable than black-box problems only when the trace feedback provides information beyond reward signals; otherwise, information-wise, OPTO is no easier than black-box problems. The Trace framework automates the generation of the execution trace in OPTO, when users of Trace decorate the workflow end-to-end. Currently, OptoPrime uses the docstring of operators in the execution trace to understand the operators; nonetheless, Trace logs also the source code of the decorated methods, which can also be used in the future to design optimizers that uses more details of the operators. In our experiments, we focus on output feedback that is automatically generated. In other contexts, e.g. users interacting with chatbots, we can natively gather natural language feedback or synthesize feedback [16] based on raw observations. We also anticipate that feedback in the form of images (e.g. users' gestures) or videos (e.g. videogame player showing a desired correction to agent's behavior) will be readily available and can be used as the output feedback in the OPTO framework. While Trace can handle this feedback, the current design of OptoPrime does not handle non-textual content (either in parameters, inputs or feedbacks). We anticipate future work along the lines of [16] on enhancing feedback design and developing guidelines for designing more informative and directive optimizer and feedback mechanisms.

**How to design adaptive optimizer for general OPTO problems?** The proposed OptoPrime optimizer shows the possibility of designing a single optimization algorithm to solve a range OPTO problems from diverse domains. However, we remark that OptoPrime is just the first step; it is akin to the vanilla gradient descent algorithm, which shows a proof of concept but is not scalable for large problems. The current design of OptoPrime has several scalability limitations due to its summary approach that updates parameters through one call to an LLM. While we show in Section 5.1, that OptoPrime can compete with ADAM in small problems, OptoPrime is not as computational efficient and cannot scale as well as ADAM; therefore, for large-scale numerical problems, Trace and OptoPrime with the current design does not replace classical AutoDiff. OptoPrime is also limited by the ability of LLMs. It has difficulty in handling parameters or nodes that cannot be compactly represented in text, which prevents it from optimizing neural network weights, or reasoning with large, stateful objects like a database. Similarly, it likely cannot handle large graphs (with thousands of nodes) at the moment, as such a large graph would result in a huge context which may be beyond what LLMs can understand and reason about reliably. This limitation is aggravated when dealing with noisy feedback or systems, as we need to present multiple graphs and feedback in the context at once. We also do not know how to rigorously define the concept of step size, which however we expect is important to handle noisy or local feedback. We need further research on graph simplification and representation, to reduce complexity and improve the efficiency of feedback propagation. Lastly, right now OptoPrime represents the constraints on parameters as part of the text description, but we have observed that LLMs do not always follow it. An effective workaround is to implement constraint checking in the workflow to throw exceptions (which are then handled as feedback). More specialized constraint handling techniques is an interesting research direction (e.g. "projecting" OPTO solver proposals for parameters onto their feasible sets), but they are not implemented in OptoPrime yet.

## B  Related Work

**Framework for Computational Workflows** Frameworks such as LangChain [29], Semantic Kernels, AutoGen [5], DSPy [22] allow composing computational workflows and provide hand-engineered optimizers to tune an LLM's context (i.e. prompt templates, few shot examples, or tool libraries) using scalar feedback with black-box search techniques. They support tracing of the workflow to aid in profiling, debugging and visualization. In contrast, Trace uses tracing for *automatic optimization*, and constructs a different representation of the computational graph which is suited for that purpose. Moreover, Trace is designed to be general-purpose and agnostic to the underlying frameworks of computational workflows users choose. In principle, one can apply Trace to decorate and tune a workflow based on a mix of Autogen, LangChain, DSPy codes. In fact, our experiments in Appendix I use workflows declared using both AutoGen and DSPy.

**Optimization of Graphs of LLM Workflows** There are multiple efforts to optimize the computational graph of LLM workflows, which is a special case of the OPTO problem. These algorithms focus on optimizing prompts. SAMMO [14] is an example for prompts that uses additional graph structure to make the optimizer efficient. SAMMO represents the prompt parameter itself as a program so as to enable more efficient black-box search through the space of programs. DSPy [22] can optimize directly the prompts or the few-shot examples to include using scalar reward feedback. Retroformer [35] uses another small language model (LM) to provide suggestions/feedback (i.e. changing prompts) to improve the behavior of an actor LLM, where the small LM is tuned by offline RL. Deep Language Networks [36] view all of the prompts in an LLM workflow as tunable parameters and jointly optimizes them. They discovered that optimizing each parameter in isolation instead produces subpar results. [37] frames LLM systems as graph where nodes are operations and edges are messages/connections. (Note that this is different from the DAG used in Trace; here nodes are messages and edges are input-output of operators) and optimizes for the connection on edges (binary variables) by REINFORCE using scalar reward feedback and prompts by LLMs . They optimize each component separately without considering each other; for example, the prompts are optimized individually without considering the graph topology or how they are used down the road. We suspect this approach can be less stable. Their prompt optimization part also does not take output feedback, but simply use an LLM to self-check whether the prompt meets the need of generating desired functions the user specified. In contrast to these works, through the OPTO framing, Trace supports *joint* optimization of all parameters (prompts, hyperparameters, codes) with *rich* feedback, and is agnostic to graph structures (e.g., changing these parameters can dynamically change the graph

structure and connections between nodes). Users of Trace are free to specify which parameters they want to automatically optimize via online interactions.

**LLM-Optimizers for Prompts and Codes**    There is a huge and fast growing literature on using LLMs as optimizers to improve prompts [12, 11, 38–40] or codes [41, 15, 42, 38]. Different from the works mentioned above, here the focus has been on an isolated problem (e.g., changing the behavior of a single LLM or improving the code generation in the question-answering format) rather than considering a non-trivial workflow or agent with multiple components like above. They do not consider optimizing prompts or codes as one component of a bigger workflow (e.g. implementing an autonomous agent), which is harder and requires the right credit assignment. In addition, these LLM-based optimizers, including OPRO [13], often propose only principles of how prompts should be designed and requires crafting problem specific prompts (as opposed to a single optimization prompt that can be applied to different problems). For adapting them to new problems, users need to design new prompts. Trace can also be applied to optimize trivial OPTO problems where the returned graph has just a single node of the parameter (which are the scenarios considered by these works). Nonetheless, the main focus of this paper is to study how optimization can be done efficiently as the graph becomes nontrivial and for diverse applications. Trace achieves this by using the abstract OPTO problem framing. Since OPTO encapsulates domain specific info in the graph, it enables designing fully instantiated optimizers that can be applied to multiple problems, rather than just principles which then requires hand crafting prompts for individual problems like previous works.

**LLM-Optimizers for Hyperparameters**    Recent works like [43, 44] use LLMs to optimize numerical hyperparameters, as an alternate to Bayesian optimization. Here in the experiments we show that Trace + OptoPrime also can effectively learn hyperparamters, faster than Bayesian optimization. The main difference between Trace and the aforementioned work is the representation of the problem. In Trace, we provide the graph to the LLM-based optimization (through the pseudo-algorithm representation), and we consume rich language feedbacks on the output, both of which accelerates hyper-parameter optimization.

**OPTO Related Setups**    OPTO is a generalization of partial monitoring games [45]. If there exists a latent loss function that the feedback $f$ adheres to (e.g. as in [23]), those OPTO instances can be written as partial monitoring game. However OPTO admits a more general notion of feedback $f$, and we discuss solution concepts for them in Appendix F. On the other hand, OPTO can be also viewed as a special case of Learning from Language Feedback (LLF) setup defined in [23] with observations as the trace feedback. This is a framing of a *meta* LLF problem. In the LLFBench Meta-World experiments of this paper (Section 5), we show Trace can be used to learn policy for LLF problems grounded to an application too.

**AutoDiff and Back-propagation**    Back-propagation has been shown to be a very effective tool in optimizing differential computational workflows. Our design of Trace is inspired by back-propagation and the ease of use of the AutoDiff framework PyTorch [19]. Nonetheless, we highlight that back-propagation (Backward Mode Differentiation) is not the only AutoDiff algorithm. For example, the gradient can be computed in a forward mode (Forward Mode Differentiation) as well, and there are also techniques of Checkpointing [46] and Truncated Back-Propagation approximation [47] for efficiency. What are the equivalent ideas of these methods for general computational workflows? We think this is an interesting future research direction.

## C    Examples of OPTO

To ground the OPTO setup, we show how OPTO is related to some existing problems with examples.

**Example 4** (Neural network with back-propagation)**.** The parameters are the weights. $g$ is the neural computational graph and $f$ is the loss. An example context $\omega$ can be "Minimize loss". The back-propagation algorithm, in view of the OPTO formulation, is embedded in the OPTO optimizer. For example, an OPTO optimizer here is a composition of back-propagation and gradient descent, where back-propagation takes $\tau$ to compute the propagated gradient at the parameter.

**Example 5** (Code Debugging)**.** The parameters are the codes. $g$ denotes the stacked trace and $f$ is the error message returned by a compiler. $\omega$ can be "Make no error".

**Example 6** (RL). The parameters are the policy. $g$ is the trajectory (of states, actions, rewards) resulting from running the policy in a Markov decision process; that is, $g$ documents the graphical model of how an action generated by the policy, applied to the transition dynamics which then returns the observation and reward, etc. $f$ can be the termination signal or a success flag. $\omega$ can be "Maximize return" or "Maximize success".

**Example 7** (Hyperparameter Tuning of ML Pipeline). The parameters are e.g. learning rates and architectures. $g$ describes the stages of the ML pipeline and the evaluation on the validation set, and $f$ is the validation loss. $\omega$ can be "Minimize validation error".

**Example 8** (Prompt Optimization of an LLM Agent). The parameters are the prompt of an LLM workflow. $g$ is the computational graph of the agent and $f$ is the feedback about the agent's behavior (which can be scores or natural language). $\omega$ can be "Maximize score" or "Follow the feedback".

**Example 9** (Multi-Agent Collaboration). The parameters are each agent's prompts. $g$ describes the entire conversation flow between agents, and $f$ is the feedback about whether the task is successful after each agent performs their action. $\omega$ can be "A group of agents coordinate to finish a task.".

As mentioned, the computational graph $g$ returned by the Trace Oracle $\mathcal{T}$ may have different graph structures. The length of the execution trace, e.g., in the debugging example above depends on how far the code executes. Similarly, the rollout length of in the RL problem can be randomly determined. The formulation of the Trace Oracle abstracts the details of a computational workflow, so problems from different domains can be framed in the unified framework. This abstraction allows us to design the computational tool Trace for various applications.

## D   Trace Handles Error in Execution as Feedback

It is worth mentioning that execution error can be directly used as feedback to optimize parameters in Trace. When execution error happens within a method decorated by bundle, Trace would adds a special exception node to the global computational graph and throw an TraceExecutionError to stop the computation. The computational graph ends at where the execution error happens. This exception node becomes the new output of the inputs to the decorated method (since the original method raises an error) and is the output of the truncated computational graph. Messages in TraceExecutionError can then be used as the feedback $f$ in OPTO and propagated from the exception node to the parameters. By calling an OPTO optimizer, the parameters can be updated to avoid causing the same execution error. See the exception handling code in Fig. 2.

We find that this error handling mechanism has two convenient usages. First, this allows using Trace to automatically debug issues in the workflow due to incorrect parameter settings. Such errors can happen frequently especially when codes are parameters, as during optimization codes not satisfying syntax or downstream API requirements can happen. The second usage is to enforce constraints the workflow has to satisfy at different stags of computation. With Trace, if an intermediate computed result does not satisfy the constraint, we can simply throw an exception which states the desired constraint. This error signal would be caught by Trace and can then provide early feedback to efficiently improve the parameters, since the graph is truncated at the error.

## E   Analysis of Trace

### E.1   Proof of Complexity

Algorithm 2 propagates the subgraph, represented by a priority queue (implemented as a min-heap). At a time, it needs to maintain the subgraphs coming from $W$ children separately. This leads to the space complexity of $O(WN)$. This $O(WN)$ space complexity leads to the extra $WN \log N$ factor in the time complexity of MSP compared with back-propagation, which is the time needed for merging $W$ subgraphs of size $O(N)$.

### E.2   Proof of Lower bounds

Consider an OPTO problem whose goal is to find a parameter matching a $k$-digit binary number. The computation checks each digit against a reference number in an arbitrary order. The feedback is either "$N^{th}$ check failed" or "All checks succeeded". Propagated feedback must communicate $k$ bits

of information to interpret the feedback correctly; and the minimal subgraph conveys exactly that information. Updating the parameter using the minimal subgraph is trivial, whereas without it there are $2^k$ possibilities to check.

# F  When is OPTO Efficiently Solvable?

We show that OPTO covers a wide range of complicated optimization problems. This shows that if OPTO can be efficiently solved, then many complex workflows can be efficiently optimized. However, the generality of OPTO also raises some fundamental questions, such as if OPTO is well defined and when OPTO can be efficiently solved. These questions stem from its generality of the context $\omega$ and the output feedback $f$ in OPTO, since e.g. they can be anything describable texts. This flexibility makes the scope of OPTO go beyond standard mathematical optimization problems, where a setup has a fixed context $\omega$ (e.g., "First-order optimization") and a fixed type of output feedback $f$ (a descent direction). Fully characterize the properties of OPTO, due to its generality, is beyond the scope of this paper and would require years of future research to come. Nonetheless, here we attempt to provide some preliminary answers and point out some research questions.

## F.1  What is a solution?

Classical mathematical optimization problems have a problem definition which itself is the solution concept. For example, in a minimization problem, it is clear we want to find the minimum of an objective function; even for problems as abstract and general as an equilibrium problem, the problem setup clearly states the solution concept of finding a point/set satisfying an equilibrium inequality [48]. One common pattern of these problems is that the solution concept is something that can be described as conditions on feedback that the parameter should satisfy.

By contrast, in a OPTO problem $(\Theta, \omega, \mathcal{T})$, by varying the context $\omega$, the desired parameter can change from one extreme to another. For example $\omega$ may state "Follow the feedback" or "The feedback is adversarial.". Therefore, we need define the solution concept of OPTO differently, rather than just using the feedback. We need to also consider the context $\omega$ appropriately. Below we make an attempt to give an axiom of OPTO for its solution to be well defined.

**Axiom 1** (Verifiability)**.**  There is an verification oracle (a human, a machine learning model, or a polynomial-time algorithm) when given $(\theta, \omega, f)$ can verify whether $\theta$ is a solution or not.

Notice the verification oracle in Axiom 1 is not limited to just algorithms. This is intentional because we currently do not have algorithms that are intelligent enough to process the wide range of contexts and feedback that OPTO allows. Therefore, we include human judgement or the use of LLMs or other AI systems as part of the definition, while acknowledging the impreciseness of the statement due to OPTO's soft computing nature. Lastly we note the verifiability is only defined with respect to the context $\omega$ and the output feedback $f$, not the execution trace $g$. That is, the verification of a solution depends only on the output of computation.

## F.2  Does a solution exist?

Under Axiom 1, we can start to ask the basic question of whether a solution to an OPTO problem exists or not. There are clearly problems where no solution exists (that is, no parameter in $\Theta$ can be verified by the verification oracle). For example if the feedback $f$ is contradicting and yet the context $\omega$ is "Follow the feedback.", then there would be no solution that is satisfactory. On the other hand, if $\omega$ is "Ignore the feedback", all parameters can be solutions. In the following, we assume solutions of OPTO under consider exist. This assumption would rule out problems, e.g., where the feedback is adversary to the context, and makes solving OPTO is a well-defined search problem.

**Assumption 1.**  For an OPTO problem $(\theta, \omega, \mathcal{T})$, we assume there is at least a parameter $\theta \in \Theta$ such that it can be verified as a solution by the verification oracle.

## F.3  Can OPTO be efficiently solved?

So far our discussion establishes OPTO as a well-defined search problem, based on qualification on the context $\omega$ and the output feedback $f$. However search problems can be NP-hard. In other words,

we know that, without the execution trace, there are search problem instances modeled by some $\omega$ and oracle giving $f$ that cannot be efficiently solved. Take RL for a tabular MDP as an example of OPTO problem. Without the execution trace (i.e., not seeing the Markovian structure and trajectories), the problem has an exponential complexity (due to the size of the policy space) and we know by using the execution trace here, tabular RL can be solved approximately in polynomial time [49]. Another example is training of neural networks. Without the execution trace, we have a complex black-box optimization with a loss value, without gradients, whereas an execution trace allows implementation of back-propagation to compute the gradients at the parameters.

More broadly speaking, if we consider a "human" as an optimizer for OPTO, we see that (expert) engineers/researchers, when equpped with additional computational tools, can efficiently solve a broad range of OPTO problems (such as by using the execution trace. From these observations, we conjecture using information in the execution trace is the key to unlock efficient OPTO. More precisely, we conjecture that OPTO is efficiently solvable when the context and the trace feedback need to provide information to construct a corrective search direction. For example, when the output feedback back is just a scalar loss, and yet the context + execution trace feedback does not provide enough information to compute a descending direction then OPTO reduces back to a black box problem. (See the problem instance in Appendix E.2). Nonetheless, identifying which subsets of OPTO are efficiently solvable is a big open research question.

# G   Additional Details of Trace and OptoPrime

## G.1   Backward Step of Trace

The MSP extracts the minimal subgrpah of the full computational graph of the workflow. Here we show a visualization using the example in Fig. 4.

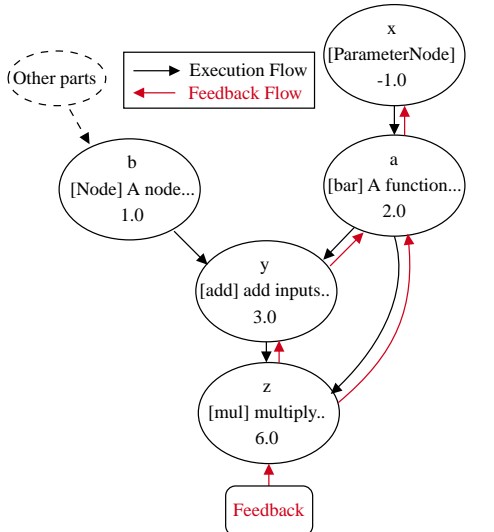

(a) This is an illustrative example of the graph constructed by Trace and how feedback is backpropagated to the parameter x.

(b) We create a succinct summary of the computation graph using a language that mimics a program.

**Figure A.1: Optimization Representation**. For a program of x = node(-1.0); a = bar(x); y = a + 1; z = a * y and the optimization objective of $\max_x z$, Trace automatically constructs a computation graph and represent the optimization problem as a debugging report. Note that the real program and the traced execution graph are different.

## G.2   Prompts used in OptoPrime

OptoPrime is an LLM-based optimizer. Its prompt is composed of the following parts.

1. System Prompt: Representation Prompt (Fig. A.2) + ReAct+CoT Output Prompt (Fig. A.3)

2. User Prompt (Fig. A.4 or Fig. A.5)

where + denotes concatenation. We list the prompt templates of different components below.

```
1   You're tasked to solve a coding/algorithm problem. You will see the instruction, the code,
     the documentation of each function used in the code, and the feedback about the execution
     result.
2
3   Specifically, a problem will be composed of the following parts:
4   - #Instruction: the instruction which describes the things you need to do or the question
     you should answer.
5   - #Code: the code defined in the problem.
6   - #Documentation: the documentation of each function used in #Code. The explanation might
     be incomplete and just contain high-level description. You can use the values in
     #Others to help infer how those functions work.
7   - #Variables: the input variables that you can change.
8   - #Constraints: the constraints or descriptions of the variables in #Variables.
9   - #Inputs: the values of other inputs to the code, which are not changeable.
10  - #Others: the intermediate values created through the code execution.
11  - #Outputs: the result of the code output.
12  - #Feedback: the feedback about the code's execution result.
13
14  In #Variables, #Inputs, #Outputs, and #Others, the format is:
15
16  <data_type> <variable_name> = <value>
17
18  If <type> is (code), it means <value> is the source code of a python code, which may
     include docstring and definitions.
19
```

**Figure A.2:** Representation Prompt that phrases the OPTO update as a pseudo-algorithm question.

```
1   Output_format: Your output should be in the following json format, satisfying the json
     syntax:
2
3   {{
4   "reasoning": <Your reasoning>,
5   "answer": <Your answer>,
6   "suggestion": {{
7       <variable_1>: <suggested_value_1>,
8       <variable_2>: <suggested_value_2>,
9       }}
10  }}
11
12  In "reasoning", explain the problem: 1. what the #Instruction means 2. what the
     #Feedback on #Output means to #Variables considering how #Variables are used in #Code
     and other values in #Documentation, #Inputs, #Others. 3. Reasoning about the suggested
     changes in #Variables (if needed) and the expected result.
13
14  If #Instruction asks for an answer, write it down in "answer".
15
16  If you need to suggest a change in the values of #Variables, write down the suggested
     values in "suggestion". Remember you can change only the values in #Variables, not
     others. When <type> of a variable is (code), you should write the new definition in the
     format of python code without syntax errors, and you should not change the function
     name or the function signature.
17
18  If no changes or answer are needed, just output TERMINATE.
19
```

**Figure A.3:** ReAct+CoT Output Prompt that instructs LLMs should respond in the format of (reasoning, answer, suggestion) and explains the output format.

```
1    Now you see problem instance:
2
3    ================================
4    {actual_problem_instance}
5    ================================
6
7    Your response:
8
```

**Figure A.4:** User Prompt for OptoPrime without Memory

```
1    Now you see problem instance:
2
3    ================================
4    {actual_problem_instance}
5    ================================
6
7    Below are some variables and their feedbacks you received in the past.
8
9    {
10        "variables": {
11            {variable1_name}: {variable1_value1}
12            {variable2_name}: {variable2_value1}
13            ...
14        },
15        "feedback": {feedback_1}
16    }
17
18    {
19        "variables": {
20            {variable1_name}: {variable1_value2}
21            {variable2_name}: {variable2_value2}
22            ...
23        },
24        "feedback": {feedback_2}
25    }
26    ...
27
28
29    Your response:
30
```

**Figure A.5:** User Prompt for OptoPrime with Memory

```
1 #Instruction
2 {instruction}
3
4 #Code
5 {code}
6
7 #Documentation
8 {documentation}
9
10 #Variables
11 {variables}
12
13 #Constraints
14 {constraints}
15
16 #Inputs
17 {inputs}
18
19 #Others
20 {others}
21
22 #Outputs
23 {outputs}
24
25 #Feedback:
26 {feedback}
```

**Figure A.6:** Problem Template used to fill the User Prompt. By default the Instruction (which is the context $\omega$ of OPTO) is "You need to change the <value> of the variables in #Variables to improve the output in accordance to #Feedback."

# H   Comparison between Trace and TextGrad

Beyond the empirical results described in Section 5.5, there are several first principles reasons to prefer the Trace design to build future LLM-powered generative optimizers.

**Joint vs. Individual Parameter Optimization:**   When there are multiple parameters to be optimized in a workflow, Trace takes a "joint optimization" view (constructing the minimal subgraph involving all of the parameters described in the initialization of OptoPrime). This flexible design also allows an alternative approach of "individual optimization" akin to co-ordinate descent if desired, i.e., fix all but one parameter and individually optimize each parameter. Such an optimization heuristic can be accomplished, e.g., by instantiating an OptoPrime instance per parameter and iterating through their updates; however, for many problems (like the toy example below), this strategy, which TextGrad employs, can be sub-optimal because a consistent optimizer must additionally maintain state about any suggested updates to all other parameters when reasoning about an individual parameter update.

```
1 @bundle()
2 def function1(x):
3     return x > 0
4 @bundle()
5 def function2(y):
6     return y % 2 == 0
7 def xor_test(x, y):
8     return function1(x).neq(function2(y))
9 input1 = node(3, trainable=True); input2 = node(4, trainable=True)
10 xor_test(input1, input2).backward(feedback="Find a set of inputs to make the return True.")
```

**Listing 1:** Individual Parameter Optimization is suboptimal in this example compared to Trace.

**Heterogenous Parameters:**   Trace encapsulates all Python datatypes as node, including numbers, floats, strings, etc. In contrast, many generative optimization libraries like TextGrad restrict their "variables" (i.e. parameters and intermediate values) to only be strings; consequently running numerical optimization experiments, or optimizing code alongwith prompts and hyper-parameters can be challenging to set up.

**Modularity:**   Trace provides a clean separation between the infrastructure (a platform akin to PyTorch) and a generative optimizer (OptoPrime akin to the Adam optimizer). Trace provides an object "TraceGraph" that describes the computational graph. This graph is independent of optimizers and LLMs. Such independent computation graph objects do not exist in TextGrad. We hope the Trace-provided interface can spur the development of many other generative optimizers, like how the OpenAI Gym API allowed the development of several Deep RL algorithms targeting a unified representation of RL problems.

**Flexible Representation:**   The bundle functionality of Trace is more flexible in representing computations to an LLM than pre-defined templates used in other optimization frameworks. TextGrad relies on special functions to chain variables together, such as using "FormattedLLMCall", while Trace supports any user-defined functions through decorators like bundle.

**Scalability:**   The strategy taken by OptoPrime can scale better to larger graphs (both in terms of error accumulation across multiple LLM calls, and the costs of multiple LLM calls) compared to requiring one LLM call per computation node (and we see with bundle that not all computation nodes present equal difficulty for an optimizer). The difference in terms of computation costs between the two strategies can be large, especially when not all operators in the graph are as complex as querying LLM. However there may be even better optimizers that blend OptoPrime and TextGrad so as to divide-and-conquer huge computation graphs, we are excited about this avenue for future research.

# I Experiment Details

## I.1 Token Counts

We include token counts for the OPRO and OptoPrime prompts used across all our experiments, at the first iteration of optimization (note that OPRO's token usages grows *linearly* with iterations). We can see that indeed OptoPrime consumes significantly more tokens than OPRO. However, we observe consistently that even allowing 7-10x more iterations of OPRO so as to equalize token costs, the OPRO performance plateaus to a worse level than OPTOPrime (e.g. Figure 1: OPRO at Iter 7 vs. OptoPrime at Iter 2; Figure 5b: OPRO at Iter 50 vs. OptoPrime at Iter 5; Figure 6b: OPRO at Iter 30 vs. OptoPrime at Iter 10, etc.). OPRO is suboptimal not due to a token limit but instead a lack of information, which is captured and represented using Trace.

| Domain | OPRO | OptoPrime |
|---|---|---|
| Numerical Opt | 175 | 918 |
| BigBench-Hard | 400 | 1883 |
| Traffic Opt | 198 | 1679 |
| MetaWorld | 470 | 7101 |
| Battleship | 437 | 1305 |

**Table A.1:** Token counts.

## I.2 Battleship

We implement a simple battleship game board in Python. The exact code is in the supplement. The game offers a string-based visualization of the board. It randomly places different types of ships on a 2-dimensional board with pre-specified width and height when it initializes. The agent does not see the ship location and has to select a coordinate on the board to hit next. One additional rule of this game is that the agent can go again if their previous coordinate selection (fire) is a hit, not counting as the finish of a turn. In Figure 1, we ran 10 trials, where in each trial, we ran 20 iterations of training. We measure the reward as % of ship squares hit (over all squares occupied by ships). The reward plateaued at 60% because the game has a chance element (heuristics and strategies can only go so far – strategy is only in effect if a hit happens. Otherwise, there is no information about where ships might be).

## I.3 Numerical Optimization

Any classical numerical optimization problem can be framed as an OPTO problem. Consider $h(x)$ and a target $y^*$, in a context $\omega$ finding the $y^*$ by changing $x$; we know the most useful corrective $f$ feedback to change $x$ is the gradient $\nabla_h x$. Similar to Trace, AutoDiff packages like PyTorch's AutoGrad have implemented dynamic graph construction with special classes like `torch.Tensor`. We want to validate whether it is possible to rely on binary text feedback, a graph automatically constructed by Trace, and OptoPrime to update $x$ in the context of minimizing $|y - y^*|$.

We constructed a synthetic task environment where we can create a complex computation graph with arbitrarily complex connections between numerical variables. The focus of this environment is on the complexity of the graph, not on the complexity of the numerical operators. Therefore, we only use one-dimensional input and basic arithmetic operators to create a numerical optimization problem solvable by a first-order optimizer. This environment constructs a computational graph by sampling a number of times. At each time, it will either use a previously computed variable or sample a new variable, and an operation will be sampled to combine them. The optimization task is, for a fixed number of steps, an optimizer needs to output $x$ that minimizes $y$.

We evaluate the following baseline methods. Basic Agent: a basic LLM agent that simply stores past information of $(x_{t-1}, y_{t-1})$ in context before choosing the next $x_t$. OPRO Agent: a basic LLM agent but we implement the state-of-the-art LLM optimizer OPRO [13], which updates the meta-prompt of the basic LLM agent. Torch + Adam: the problem we construct is end-to-end differentiable. Therefore, we simply pass in `torch.Tensor(x)` as input and use Adam optimizer to update. We tune the learning rate slightly and found 1e-1 to work well. We compare two kinds of Trace-based

optimizers: Trace, where we allow OptoPrime to read in the entire computation graph before updating $x$, or Trace Masked, where we hide the computation graph.

We run 30 trials over different computation graphs and start all methods with the same initial $x, y^*$. We compute the absolute error, which is $|y - y^*|$. On average, Trace is able to match the best-in-class first-order gradient optimizer Adam [25]. It is not entirely surprising that all the other baselines are performing worse due to a lack of access to the computation graph. To our surprise, OPRO, by only accessing the history of input and output, as well as changing the meta-prompt, is able to eventually discover the correct solution. This confirms why there were early signs of success using LLMs for black-box optimization in a simple plug-and-play style. However, OPRO is not an efficient optimizer because it lacks access to the Trace oracle. We show OPRO struggles even more when the computation graph gets more complex.

## I.4 Traffic Control

We tested OptoPrime in a traffic control problem which is an instance of hyper-parameter tuning. We used UXSim [26] to simulate traffic at a four-way intersection, where the tunable parameters are the duration of the green lights for each direction of traffic flow. The feedback is a scalar loss calculated by monitoring the flow of a pseudo-random sequence of vehicles arriving at the intersection over a period of 30 minutes. The loss computes an estimate of the delay experienced by all vehicles due to the intersection, as well as variability in this estimate for every link in the network; lower values are better. The goal of an optimizer is to identify values for all of the green light duration so as to minimize the loss using the fewest number of traffic simulations. If the green light duration for a given traffic flow direction is set too low, then vehicles will queue up over time and experience delays, thereby lowering the score for the intersection. However, if the green light duration for a given direction is set too high, vehicles in other directions will queue up and experience delays, thereby lowering the score for the intersection. Hence an optimizer must find the right trade-off for temporally distributed and variable demands.

In Figure 5 we report the performance of a SOTA heuristic from the traffic control literature, SCATS [27] (adapted to this toy setting) as well as two black-box optimization techniques: Gaussian Process Minimization (GP) [8] and Particle Swarm Optimization (PSO) [28]. All methods are initialized to evaluate the same starting parameter. GP and PSO further evaluate 5 random parameters; moreover, if they query a previously evaluated point, that query is replaced by a randomly sampled parameter. GP constructs a surrogate model to mimic the black-box traffic simulation function which maps from parameters to observed score. Then it minimizes a utility function (e.g. the lower confidence bound) using the surrogate model to pick the next parameter to evaluate. PSO on the other hand maintains 5 particles in parameter space, each with a position and velocity. At each iteration of PSO, particles update their positions according to their previous positions and velocity, evaluate the function at the updated positions, and update the velocities of all particles using the observed values. Although GP and PSO are both black-box methods, GP can be thought to replace Trace oracle with instead a smooth differentiable surrogate function; whereas PSO is very different and maintains a candidate set of parameters (can be thought of as conceptually related to OptoPrime with memory).

GP appears to be bad because even when it converged, the exploration heuristic randomly samples parameters rather than pick the converged parameter. PSO appears bad because 10 iterations is insufficient for its convergence. Note that given enough number of iterations, black-box approaches will eventually perform well. Trace is quickly competitive with the SCATS heuristic, whereas OPRO is not. Moreover, we find that memory is crucial for OptoPrime to perform well at this task. Finally, Trace consumes additional overhead compared to black-box methods; beyond the space and time complexity for running the traffic simulation, Trace additionally materializes the computation graph per iteration. Thus it can also be more expensive per LLM call compared to OPRO.

## I.5 BigBench-Hard

Perhaps more surprisingly, there are many components that a workflow needs to learn. Some of these components can be the prompt to generate output from an LLM, while other components can be code that needs to further process these outputs. In many workflows today, enabled by LangChain [29] and DSPy [22], only a small part of this workflow, predominantly, the input to an LLM API call, is optimized. These libraries optimize input to an LLM, and human engineers process that input

```
1  import trace
2
3  class Predict(trace.Module):
4      def __init__(self):
5          self.prompt_template = trace.node("""
6          Given `question`, produce the `answer`.
7          question: {}
8          answer:
9          """,
10         trainable=True)
11
12     def forward(self, question):
13         user_prompt = self.create_prompt(question)
14         response = self.call_llm(user_prompt)
15         answer = self.extract_answer(question,
    response)
16         return answer
17
18     @trace.bundle(trainable=True)
19     def create_prompt(self, question):
20         """formulate the prompt with the question"""
21         return self.prompt_template.format(question)
22
23     @trace.bundle(trainable=True)
24     def extract_answer(self, question, response):
25         """Extract the answer out of LLM response"""
26         answer = response.split("answer:")[1].strip()
27         return answer
```

(a) We write a workflow that prompts an LLM for a question and extracts the answer.

```
1  import trace
2
3  dataset = load_dataset(link, task)
4  predict = Predict()
5  optimizer =
       trace.Optimizer(predict.parameters())
6
7  for ex in dataset:
8      # Forward pass
9      response = dp.forward(ex['question'])
10     if response == ex['answer']:
11         feedback = "The answer is correct"
12     else:
13         feedback = "The answer is incorrect"
14
15     # Backward pass and update
16     optimizer.zero_feedback()
17     optimizer.backward(response, feedback)
18     optimizer.step()
```

(b) The optimizer class takes in any parameter regardless of whether it is code or text. Although the actual optimization implementation can provide different treatments to many input types, the user interface stays consistent.

**Figure A.7: LLM-based Workflow Optimization Example.**

and integrate it into other systems. Indeed, both libraries can enable robust and swift large-scale engineering efforts to build LLM-based software. However, if our goal is to develop self-adapting agents that can modify their own behavior, we should not ignore one of LLM's greatest strengths: code generation. Trace allows us to unify prompt optimization and code generation, which enables the creation of agents capable of fast learning.

In this example of an LLM-based workflow (Figure A.7), there are three parameters that are flagged as trainable for the optimizer: prompt_template, create_prompt, and extract_answer. Note that two of them require the LLM to generate Python code, and one of them requires the LLM to modify a text. Trace abstracted away the different data types and enabled direct update and optimization of them. Furthermore, a human engineer is often tasked with writing an error-free extract_answer. The output of an LLM can be highly stochastic and can often change over time; the code that is used to extract the response of an LLM has to be extremely robust and, therefore, arduous to create. Whenever a major distribution shift happens in the LLM output, this code needs to be rewritten by a human engineer, and it is hard for humans to anticipate all of LLM's output patterns.

We set up the task of end-to-end workflow optimization. Unlike a typical LLM benchmark evaluation, where a lot of effort went into creating the perfect evaluate(answer, target) method so that all kinds of LLM outputs were post-processed, cleaned, and formatted to match the ground truth, we choose a simple evaluation function (that extracts a segment or does exact string matching) and place the burden on the workflow itself to figure out how to create the right answer to satisfy the evaluation metric. We choose Big-Bench Hard [21] as our task because it has 23 subtasks and contains both language and algorithmic tasks.

We split each task dataset into training, validation, and test. For Trace and Trace-CoT, we use the first 15 examples for training, 5 examples for validation (picking the best learned workflow), and then evaluate the performance on test examples. We use template-based positive and negative feedback during training. The positive feedback is "The answer is correct! No need to change anything." The negative feedback is "The answer is wrong. We expect the output of your answer to be {ANSWER}. Please modify the prompt and relevant parts of the program to help LLM produce the right answer." DSPy's prompt optimization method does not explicitly require a validation set, therefore, we just used all 20 examples for training. For both, we only optimize for 1 epoch. We either start with the vanilla boilerplate prompt template used by DSPy or we use the slightly sophisticated template used by DSPy's CoT module. Trace optimizes both DSPy's original design and outperforms their own optimizer COPRO by 10% on algorithmic tasks.

Big-Bench Hard requires different answer outputs. Out of 23 tasks, 14 tasks require a multiple-choice answer with options provided in the question. 4 of them require yes/no. 1 task requires True/False, while 1 task requires valid/invalid. And the 3 remaining tasks require answers that contain words or

| Task Name | DSPy | +CoT | DSPy-PO | +CoT | Trace | +CoT |
|---|---|---|---|---|---|---|
| tracking shuffled objects 7 objects | 37.39 | 90.0 | 90.43 | 90.43 | 37.8 | 87.8 |
| salient translation error detection | 51.3 | 70.87 | 51.3 | 69.57 | 63.0 | 70.0 |
| tracking shuffled objects 3 objects | 39.13 | 94.35 | 97.39 | 93.91 | 38.7 | 96.5 |
| geometric shapes | 50.43 | 62.17 | 59.13 | 60.43 | 49.6 | 62.2 |
| object counting | 0.0 | 40.0 | 0.0 | 74.35 | 42.2 | 80.4 |
| word sorting | 0.0 | 0.0 | 0.0 | 0.43 | 84.3 | 74.3 |
| logical deduction five objects | 70.0 | 80.43 | 70.0 | 74.78 | 48.7 | 75.7 |
| hyperbaton | 74.78 | 86.52 | 74.78 | 88.26 | 78.3 | 91.7 |
| sports understanding | 0.0 | 0.0 | 0.0 | 0.0 | 79.6 | 45.7 |
| logical deduction seven objects | 68.7 | 64.78 | 68.7 | 64.78 | 45.7 | 69.6 |
| multistep arithmetic two | 0.0 | 93.04 | 0.0 | 93.04 | 94.8 | 88.7 |
| ruin names | 84.35 | 87.83 | 84.35 | 87.83 | 87.8 | 90.0 |
| causal judgement | 7.78 | 70.66 | 74.25 | 70.66 | 70.1 | 54.5 |
| logical deduction three objects | 85.22 | 97.39 | 85.22 | 97.83 | 91.7 | 97.0 |
| formal fallacies | 1.74 | 81.3 | 62.17 | 81.3 | 73.5 | 67.8 |
| snarks | 86.08 | 87.34 | 86.08 | 87.97 | 81.6 | 87.3 |
| boolean expressions | 0.0 | 98.26 | 64.35 | 98.26 | 88.7 | 96.5 |
| reasoning about colored objects | 53.04 | 91.3 | 89.13 | 91.3 | 91.3 | 95.7 |
| dyck languages | 0.0 | 8.7 | 7.83 | 8.7 | 26.5 | 9.6 |
| navigate | 0.0 | 95.65 | 0.0 | 97.39 | 59.6 | 92.6 |
| disambiguation qa | 67.83 | 66.09 | 73.91 | 66.09 | 75.7 | 59.1 |
| temporal sequences | 99.57 | 99.13 | 97.39 | 99.13 | 97.8 | 98.3 |
| web of lies | 0.0 | 0.0 | 0.0 | 0.0 | 49.6 | 90.4 |
| tracking shuffled 5 objects | 37.83 | 96.09 | 37.83 | 96.09 | 58.3 | 88.7 |
| penguins in a table | 69.84 | 92.86 | 97.62 | 92.86 | 81.7 | 91.3 |
| movie recommendation | 83.48 | 76.09 | 83.48 | 76.09 | 81.3 | 75.7 |
| date understanding | 69.13 | 85.65 | 69.13 | 85.65 | 70.4 | 85.7 |

**Table A.2:** Big Bench-Hard Per-Task Result. 0-shot performance. Some 0.0 here shown is because DSPy cannot find the clean/stripped output that matches what the automatic evaluation method expects. With additional human engineering, these numbers can improve.

numbers. Even though DSPy's meta-prompt optimization is trained on each task individually, the output of LLM to the evaluation method is still not post-processed, resulting in low performances of these tasks. However, Trace can optimize code and LLM prompt jointly to successfully deliver the response expected by an automatic evaluation method.

## I.6  LLFBench Meta-World

We test the ability of Trace to optimize long-horizon workflows with complex dependencies. We experiment with using Trace to train controller (python code) for a simulated Sawyer robot manipulator. We use the Meta-World environment of LLF-Bench [23] as the simulator and consider three tasks reach, pick-place and push. LLF-Bench is a simulated benchmark with gym interface for testing an agent's ability to learn from language feedback. In these LLF-Bench Meta-World tasks, the observation is a dictionary where each field denotes a feature of the state and has a vector value (e.g., the end-effector position, the goal position, the gripper status, etc.). The keys of the observation dictionary can differ for each task. The action space is 4-dimensional, which controls the relative position of the end-effector and the state of the gripper. In each time step, the LLF-Bench Meta-World simulator returns the observation dictionary and natural language feedback to guide the robot (we use the 'a' mode of LLF-Bench, with which the language feedback would contain information about the current performance, explanation of past successes and failures, and suggestions for the next step). An episode ends if the robot successfully solve the problem or because of time-out. For each task, LLF-Bench also provides a task instruction explaining that the task is about controlling a Sawyer robot arm and the meaning of the action space (see [23]). We use that as the context $\omega$ of the OPTO problem. We consider an episodic setting. For each experiment (a random seed), we randomly sample an initial configuration. Then for each iteration of optimization, we reset the simulator to that sampled

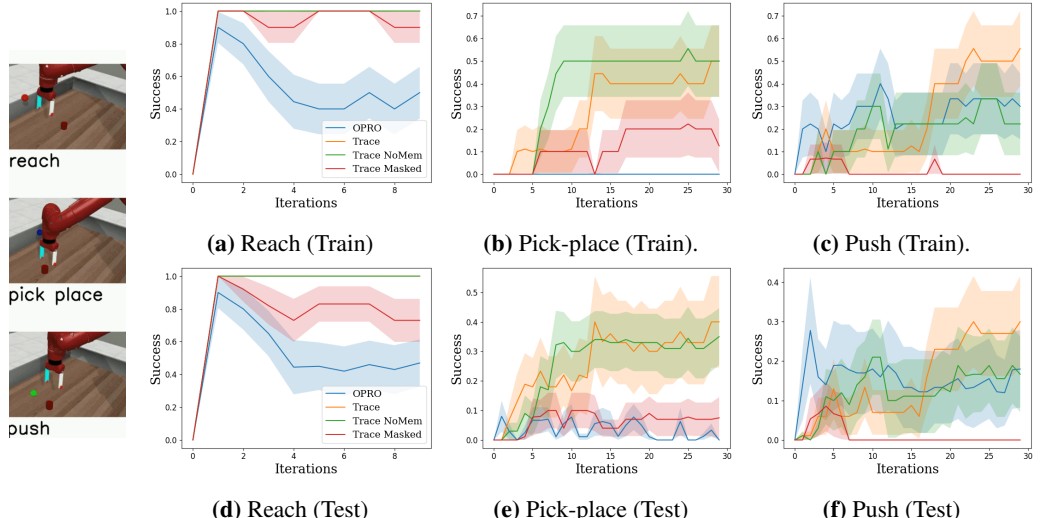

**(a)** Reach (Train)     **(b)** Pick-place (Train).     **(c)** Push (Train).

**(d)** Reach (Test)     **(e)** Pick-place (Test)     **(f)** Push (Test)

**Figure A.8:** Learning the feedback control policy (code) for a simulated Sawyer manipulator in LLF-Bench Metawrold. In each iteration (x-axis), one episode of rollout is performed and then the policy is updated. Mean and standard error of success rate over 10 seeds are shown.

initial configuration and run the robot policy for $10^{11}$ steps or until the episode termination due to success. We compute the sum of rewards and gives the output feedback $f$ in texts in the format of "Success: <true/false> Return: <score>". Note that the initial condition for all iterations within an experiment is the same so that the optimization problem is deterministic. To evaluate the learned policy's performance, for each experiment, we additionally run the learned policy starting from 10 held-out initial conditions, different from the fixed training initial condition. For each training algorithm discussed, we run it with 30 iterations, where each iteration consists of one episode rollout and one update.

To optimize the controller with Trace, we declare the control code as the parameter using the `bundle` decorator with `trainable` set to True; the initial control code simply outputs a zero vector $[0, 0, 0, 0]$. We decorate also the reset and the step function of the gym environment, so that the entire rollout of an episode can be traced end-to-end. In our implementation, a prototypical rollout would create a graph with around 30 operations where the controller code parameter is used multiple times. This graph structure is similar to that of running a recurrent neural network. For Trace, we experiment with OptoPrime with and without a memory of size 10. In addition to Trace, we implement the state-of-the-art LLM optimizer OPRO [13] as a baseline. Compared with Trace, OPRO does not use the execution trace information but rely on just memory of parameter and feedback pairs[12] To make the comparison with Trace fair, we append the feedback observation from LLF-Bench given at each time step to the final feedback received by OPRO; on the other hand, Trace uses the simple final feedback of success and return and has to read the per-step feedback from the execution trace. To run OPRO in the OPTO setting, our implementation only proposes a single candidate in each iteration, which is then evaluated and provided with the output feedback. Since in [13] OPRO generates about 10 samples per iteration, so one iteration in [13] is roughly equivalent to 10 iterations here.

The experimental results are summarized in Fig. A.8, where we show the success rates at both the training initial condition as well as the held-out testing initial conditions over 10 seeds. OptoPrime is clearly the top-performing optimizer, especially the version with memory. OPRO is able to solve Reach at the start but its performance degraded over iterations (this instability was observed in [13]) and gets similar performance as OptoPrime (without memory) in Push. To validate that the performance of OptoPrime is indeed due to using the execution trace, we include an ablation where we mask out information in #Inputs, #Others, #Code, #Definition in the LLM context (see Fig. A.1b),

---

[11]We set the problem horizon to be 10 steps, as we find the expert policies implemented in LLF-Bench can solve these problems within 10 steps.

[12]The original version of OPRO uses parameter-score pairs. Since we're interested in the more general setup of OPTO, we extend it to use parameter-feedback pairs.

which lead to significant degrade in performance and stability. This ablation shows that additionally using the execution trace provides more informed search direction compared with just using just the output feedback, which agrees with our hypothesis.

This experiment features the most complex graph structures, and using Trace for optimization here is similar to back-propagation over time. The experimental results here are quite impressive, showing that Trace is able to learn a complex control logic in a dozens of interactions, not only working on the training initial conditions but also on the held-out testing ones too. Nonetheless, we want to point out some limitations in the current experimental results. We find that the success rate of the learned policy varies largely across random seeds. Except for Reach (the simplest task), in a seed, often either it finds a policy close to 1.0 success rate or 0.0 success rate. Therefore, the plots can roughly be interpreted as how long it takes to find a working policy. In addition, in these experiments, we find that providing task-related context is necessary. We find the context needs to be informative enough for humans to understand the problem[13]; otherwise, the optimization can be solved efficiently with the time scale considered here. Nonetheless, this requirement is reasonable, as there is no free lunch.

---

[13]The original instructions in the v2 environments of LLF-Bench does not contain task specific background, but only the task name. We find this task name alone, except for Reach, is too vague even for humans to understand how to read the observation dictionary is related to the problem, so we added additionally one line to explain the task. Pick-place: "The goal of the task is to pick up a puck and put it to a goal position.". Push: "The goal of the task is to push a puck to a goal position.". Without this extra information, none of the LLM-based optimizers works in the experiments.

# J Examples of the Optimized Parameters in the Experiments

```python
class Policy(trace.Module):
    def init(self, width, height):
        pass

    def __call__(self, map):
        return self.select_coordinate(map).data

    def select_coordinate(self, map):
        plan = self.reason(map)

        act = self.act
        output = act(map, plan)
        return output

    @trace.bundle(trainable=True)
    def act(self, map, plan):
        """
        Given a map, select a target coordinate in a Battleship game. In map, O denotes
        misses, X denotes successes, and . denotes unknown positions.
        """
        return

    @trace.bundle(trainable=True)
    def reason(self, map) -> str:
        """
        Given a map, analyze the board in a Battleship game. In map, O denotes misses, X
        denotes successes, and . denotes unknown positions.
        """
        return [0, 0]

def train(epochs=10):
    policy = Policy()
    optimizer = FunctionOptimizerV2Memory(policy.parameters())
    board = BattleshipBoard()

    feedback = ""
    rewards = []
    board = board.reset()
    obs = node(board.get_shots())  # init observation
    i = 0
    while i < epochs:
        try:
            output = policy.select_coordinate(obs)
            # not traced
            obs, reward, terminal, feedback = user_fb_for_placing_shot(board, output.data)
        except TraceExecutionError as e:  # this is a retry
            output = e.exception_node
            feedback = output.data
            reward, terminal = 0, False

        if terminal:
            board = reset_board()
            obs = node(board.get_shots())  # init observation

        # Update
        optimizer.zero_feedback()
        optimizer.backward(output, feedback)
        optimizer.step(verbose=True)
```

**Figure A.9:** Battleship Agent and optimization code.

```
1  ## Iteration 1 ; Success : False
2  def reason(self, map) -> str:
3      """
4      Given a map, analyze the board in a game. On map, O denotes misses, X denotes successes,
        and . denotes unknown positions.
5      """
6      # Check for first unknown position to try as a potential move
7      for row_index, row in enumerate(map):
8          for col_index, val in enumerate(row):
9              if val == '.':
10                 return [row_index, col_index]
11     # Default to [0, 0] if no unknown positions found
12     return [0, 0]
13
14 ## Iteration 2 ; Success : True
15 def reason(self, map) -> str:
16     """
17     Given a map, analyze the board in a game. On map, O denotes misses, X denotes successes,
        and . denotes unknown positions.
18     """
19     # Improved strategic approach by checking . surrounded by more . which might indicate
        undetected area
20     max_surrounding = -1
21     best_spot = [0, 0]
22     for row_index, row in enumerate(map):
23         for col_index, val in enumerate(row):
24             if val == '.':
25                 surrounding = 0
26                 for di in [-1, 0, 1]:
27                     for dj in [-1, 0, 1]:
28                         ni, nj = row_index + di, col_index + dj
29                         if 0 <= ni < len(map) and 0 <= nj < len(map[0]) and map[ni][nj] ==
        '.':
30                             surrounding += 1
31                 if surrounding > max_surrounding:
32                     max_surrounding = surrounding
33                     best_spot = [row_index, col_index]
34     return best_spot
35
36 ## Iteration 8 ; Success : True
37 def reason(self, map) -> str:
38     """
39     Given a map, analyze the board in a game. On map, O denotes misses, X denotes successes,
        and . denotes unknown positions.
40     """
41     max_surrounding = -1
42     best_spot = [0, 0]
43     for row_index, row in enumerate(map):
44         for col_index, val in enumerate(row):
45             if val == '.':
46                 surrounding = 0
47                 for di in [-1, 0, 1]:
48                     for dj in [-1, 0, 1]:
49                         ni, nj = row_index + di, col_index + dj
50                         if 0 <= ni < len(map) and 0 <= nj < len(map[0]):
51                             if map[ni][nj] == '.':
52                                 surrounding += 1
53                             elif map[ni][nj] == 'X':
54                                 surrounding += 1.5  # increasing emphasis on proximity to
        successful hits
55                 if surrounding > max_surrounding:
56                     max_surrounding = surrounding
57                     best_spot = [row_index, col_index]
58     return best_spot
59
```

**Figure A.10:** Learned Battleship Code (reason). Functions with the same name are learned during different iterations or trials.

```
1 ## Iteration 0 (Initialization); Success : False
2
3
4 ## Iteration 1 ; Success : False
5 def act(self, map, plan):
6     """
7     Given a map, select a target coordinate in a game. On map, O denotes misses, X denotes
        successes, and . denotes unknown positions.
8     """
9     # Example implementation, more sophisticated logic may be required based on intended
        gameplay mechanics
10    return [0, 1]
11
12 ## Iteration 2 ; Success : True
13 def act(self, map, plan):
14    """
15    Given a map, select a target coordinate in a game. On map, O denotes misses, X denotes
        successes, and . denotes unknown positions.
16    """
17    # Use information from plan, which is output from __code2, checking strategic position,
        not just first available
18    best_score = -1
19    best_move = [0, 0, 0]  # initially no move is considered
20    for row_index, row in enumerate(map):
21        for col_index, val in enumerate(row):
22            if val == '.':  # consider all unknown spots
23                hopes_surrounding = 0  # will hold number of '.'s around the spot
24                for di in [-1, 0, 1]:
25                    for dj in [-1, 0, 1]:
26                        ni, nj = row_index + di, col_index + dj
27                        if 0 <= ni < len(map) and 0 <= nj < len(map[0]):
28                            if map[ni][nj] == '.':
29                                hopes_surrounding += 1
30                score = hopes_surrounding  # more '.' around, higher the chance of hit
31                if score > best_score:
32                    best_score = score
33                    best_move = [row_index, col_index]
34    if map[plan[0]][plan[1]] == '.' and best_score == -1:
35        return plan  # if no better move found, and original place is still unknown
36    return best_move if best_score > -1 else plan  # return the best move found, or stick
        with the plan if still valid
37
38 ## Iteration 8 ; Success : True
39 def act(self, map, plan):
40    """
41    Given a map, select a target coordinate in a game. On map, O denotes misses, X denotes
        successes, and . denotes unknown positions.
42    """
43    best_score = -1
44    best_move = [0, 0]
45    for row_index, row in enumerate(map):
46        for col_index, val in enumerate(row):
47            if val == '.' and [row_index, col_index] != plan:
48                hopes_surrounding = 0
49                for di in [-1, 0, 1]:
50                    for dj in [-1, 0, 1]:
51                        ni, nj = row_index + di, col_index + dj
52                        if 0 <= ni < len(map) and 0 <= nj < len(map[0]):
53                            if map[ni][nj] == '.':
54                                hopes_surrounding += 1  # favoring positions with more
    unknowns surrounding
55                            elif map[ni][nj] == 'X':
56                                hopes_surrounding += 2  # increased incentive for moves near
    successful spots
57                score = hopes_surrounding
58                if score > best_score:
59                    best_score = score
60                    best_move = [row_index, col_index]
61    if best_score > -1:
62        return best_move
63    return plan
64
```

**Figure A.11:** Learned Battleship Code (act). Functions with the same name are learned during different iterations or trials.

```python
@trace_class
class Predict(LLMCallable):
    def __init__(self):
        super().__init__()

        self.demos = []
        self.prompt_template = dedent("""
        Given the fields 'question', produce the fields 'answer'.

        ---

        Follow the following format.

        Question:
        Answer:

        ---
        Question: {}
        Answer:
        """)

        self.prompt_template = trace.node(self.prompt_template, trainable=True,
                                          description="[ParameterNode] This is the Prompt
     Template to the LLM...")

    @trace.bundle(trainable=True)
    def extract_answer(self, prompt_template, question, response):
        """
        Need to read in the response, which can contain additional thought, delibration and
     an answer.
        Use code to process the response and find where the answer is.
        Can use self.call_llm("Return the answer from this text: " + response) again to
     refine the answer if necessary.

        Args:
            prompt_template: The prompt that was used to query LLM to get the response
            question: Question has a text describing the question but also "Options"
            response: LLM returned a string response
                      Process it and return the answer in the exact format that the
     evaluator wants to see.
                      Be mindful of the type of answer you need to produce.
                      It can be (A)/(B), a number like 8, or a string, or Yes/No.
        """
        answer = response.split("Answer:")[1].strip()
        return answer

    @trace.bundle(trainable=True)
    def create_prompt(self, prompt_template, question):
        """
        The function takes in a question and then add to the prompt for LLM to answer.
        Args:
            prompt_template: some guidance/hints/suggestions for LLM
            question: the question for the LLM to answer
        """
        return prompt_template.format(question)

    def forward(self, question):
        """
        question: text

        We read in a question and produces a response
        """
        user_prompt = self.create_prompt(self.prompt_template, question)
        response = self.call_llm(user_prompt)
        answer = self.extract_answer(self.prompt_template, question, response)
        return answer
```

**Figure A.12:** Starting Code for BigBench. We write it in a similar style to DSPy's Predict module.

```
1  @trace_class
2  class PredictCoT(LLMCallable):
3      def __init__(self):
4          super().__init__()
5
6          self.demos = []
7          self.prompt_template = dedent("""
8          Given the fields 'question', produce the fields 'answer'.
9
10         ---
11
12         Follow the following format.
13
14         Question: question
15         Reasoning: Let's think step by step in order to produce the answer. We ...
16         Answer: answer
17
18         ---
19         Question: {}
20         """)
21
22         self.prompt_template = trace.node(self.prompt_template, trainable=True,
23                                           description="[ParameterNode] This is the Prompt
    Template to the LLM...")
24
25     @trace.bundle(trainable=True)
26     def extract_answer(self, prompt_template, question, response):
27         """
28         Need to read in the response, which can contain additional thought, deliberation and
    an answer.
29         Use code to process the response and find where the answer is.
30         Can use self.call_llm("Return the answer from this text: " + response) again to
    refine the answer if necessary.
31
32         Args:
33             response: LLM returned a string response
34                       Process it and return the answer in the exact format that the
    evaluator wants to see.
35                       Be mindful of the type of answer you need to produce.
36                       It can be (A)/(B), a number like 8, or a string, or Yes/No.
37             question: Question has a text describing the question but also "Options"
38         """
39         answer = response.split("Answer:")[1].strip()
40         return answer
41
42     @trace.bundle(trainable=True)
43     def create_prompt(self, prompt_template, question):
44         """
45         The function takes in a question and then add to the prompt for LLM to answer.
46         The prompt should instruct the LLM to reason, think.
47         Args:
48             prompt_template: some guidance/hints/suggestions for LLM
49             question: the question for the LLM to answer
50         """
51         return prompt_template.format(question)
52
53     def forward(self, question):
54         """
55         question: text
56
57         We read in a question and produces a resposne
58         """
59         user_prompt = self.create_prompt(self.prompt_template, question)
60         response = self.call_llm(user_prompt)
61         answer = self.extract_answer(self.prompt_template, question, response)
62         return answer
63
```

**Figure A.13:** Starting Code for BigBench. We write it in a similar style to DSPy's Predict CoT (0-shot Chain-of-Thought) module.

```
1 ## Iteration 0 ( initialization )
2 def create_prompt(self, prompt_template, question):
3     """
4     The function takes in a question and then add to the prompt for LLM to answer.
5     Args:
6         prompt_template: some guidance/hints/suggestions for LLM
7         question: the question for the LLM to answer
8     """
9     return prompt_template.format(question)
10
11 ## Iteration > 0
12 def create_prompt(self, prompt_template, question):
13     """
14     The function takes in a question and then add to the prompt for LLM to answer.
15     The prompt should now further instruct the LLM to carefully track the ball swaps
         occurring step-by-step.
16     Args:
17         prompt_template: some guidance/hints/suggestions for LLM
18         question: the question for the LLM to answer
19     """
20     prompt_template = 'Process this carefully: Step-by-step.' + prompt_template
21     return prompt_template.format(question)
22
```

**Figure A.14:** Learned Predict module for BigBench. Functions with the same name are learned during different iterations or trials.

```python
## Iteration 0 ( initialization )
def extract_answer(self, prompt_template, question, response):
    """
    Need to read in the response, which can contain additional thought, delibration and an
     answer.
    Use code to process the response and find where the answer is.
    Can use self.call_llm("Return the answer from this text: " + response) again to refine
     the answer if necessary.

    Args:
        prompt_template: The prompt that was used to query LLM to get the response
        question: Question has a text describing the question but also "Options"
        response: LLM returned a string response
                Process it and return the answer in the exact format that the evaluator
     wants to see.
                Be mindful of the type of answer you need to produce.
                It can be (A)/(B), a number like 8, or a string, or Yes/No.
    """
    answer = response.split("Answer:")[1].strip()
    return answer

## Iteration > 0
def extract_answer(self, response):
    """
    Need to read in the response, which can contain additional thought, deliberation and an
     answer.
    Use code to process the response and find where the answer is.
    Can use self.call_llm("Return the answer from this text: " + response) again to refine
     the answer if necessary.
    Args:
        response: LLM returned a string response
                Process it and return the answer in the exact format that the evaluator
     wants to see.
                Be mindful of the type of answer you need to produce.
                It can be (A)/(B), a number like 8, or a string, or Yes/No.
        question: Question has a text describing the question but also "Options"
    """
    answer = ''
    segments = response.split('\n')
    for segment in segments:
        if 'Answer:' in segment:
            answer = segment.split('Answer:')[1].strip()
    refined_answer = self.call_llm('Return the refined answer from this text: ' + answer)
    return refined_answer

def extract_answer(self, prompt_template, question, response):
    """
    Processes the LLM response and extracts the final answer in the required format.
    """
    # Assuming that the relevant part of the response is after 'Answer:' and before any
     further commentary
    extracted_part = response.split('Answer: ')[1].split(' ')[0].strip()
    # Find the section of the answer and return it directly
    result = re.search('\([A-E]\)', extracted_part)
    if result:
        return result.group()
    else:
        return 'No valid answer found'

def extract_answer(self, prompt_template, question, response):
    """
    Processes the LLM response, extracting and formatting the final answer.
    Uses code to meticulously parse the response to locate the answer section.
    Optionally refines the answer by querying the LLM again if necessary.

    Args:
        response: string from LLM, expected format contains 'Answer:' followed by the answer.
        question: Description of the question being addressed, may include 'Options'
    """
    answer = response.split('Answer:')[1].split('.')[0].strip().lower().split(',')[0].strip()
    return answer)
```

**Figure A.15:** Learned Predict module for BigBench. Functions with the same name are learned during different iterations or trials.

```
1  ## Iteration 0 ( initialization )
2  prompt_template = """
3  Given the fields 'question', produce the fields 'answer'.
4
5  ---
6
7  Follow the following format.
8
9  Question:
10 Answer:
11
12 ---
13 Question: {}
14 Answer:
15 """
16
17 ## Iteration > 0
18 prompt_template = """
19 Given the fields 'question', produce the fields 'answer'.
20
21 ---
22
23 Follow the following format.
24
25 Question:
26 Answer: Please respond precisely in the format required.
27
28 ---
29 Question: {}
30 """
31
```

**Figure A.16:** Learned BigBench Prompt Template.

```
1
2 ## Iteration 0 (initialization); Success: False
3 def controller(obs):
4     """
5     A feedback controller that computes the action based on the observation.
6
7     Args:
8         obs: (dict) The observation from the environment. Each key is a string (indicating a
     type of observation) and the value is a list of floats.
9     Output:
10        action: (list or nd.array) A 4-dimensional vector.
11     """
12     return [0, 0, 0, 0]
13
14 ## Iteration 1; Success: False
15 def controller(obs):
16     """
17     A feedback controller that computes the action based on the observation.
18
19     Args:
20         obs: (dict) The observation from the environment. Each key is a string (indicating a
     type of observation) and the value is a list of floats.
21     Output:
22        action: (list or nd.array) A 4-dimensional vector.
23     """
24     hand_pos = obs['hand_pos']
25     goal_pos = obs['goal_pos']
26     action = [goal_pos[0] - hand_pos[0], goal_pos[1] - hand_pos[1], goal_pos[2] -
     hand_pos[2], 0]
27     # Adjusting the last element of the action based on whether the hand is close enough to
     trigger grasping
28     if abs(goal_pos[0] - hand_pos[0]) < 0.02 and abs(goal_pos[1] - hand_pos[1]) < 0.02 and
     abs(goal_pos[2] - hand_pos[2]) < 0.02:
29         action[3] = 1 # Assuming action[3] = 1 triggers closing the gripper
30     return action
31
32 ## Interation 5; Success: False
33 def controller(obs):
34     """
35     A feedback controller that computes the action based on the observation.
36
37     Args:
38         obs: (dict) The observation from the environment.
39     Output:
40        action: (list or nd.array) A 4-dimensional vector.
41     """
42     hand_pos = obs['hand_pos']
43     puck_pos = obs['puck_pos']
44     goal_pos = obs['goal_pos']
45     gripper_open = 1 if obs['gripper_distance_apart'][0] > 0.5 else 0
46
47     # Adjusting strategy to close the gripper when in proximity of the puck
48     if gripper_open and ((abs(hand_pos[0] - puck_pos[0]) < 0.05) and (abs(hand_pos[1] -
     puck_pos[1]) < 0.05) and (abs(hand_pos[2] - puck_pos[2]) < 0.05)):
49         action = [0, 0, 0, 1]  # Close the gripper
50     elif not gripper_open:
51         direction_to_goal = [0.09 - hand_pos[0], 0.95 - hand_pos[1], 0.12 - hand_pos[2]]
52         action = [direction_to_goal[0], direction_to_goal[1], direction_to_goal[2], 0]  #
     Move towards the suggested pose once puck is grasped
53     else:
54         # Move towards the puck first if not carrying it
55         direction_to_puck = [puck_pos[0] - hand_pos[0], puck_pos[1] - hand_pos[1],
     puck_pos[2] - hand_pos[2]]
56         action = [direction_to_puck[0], direction_to_puck[1], direction_to_puck[2], 0]
57
58     return action
```

**Figure A.17:** Learned Code for LLFBench Meta-World Pick-Place (Part 1). Functions with the same name are learned during different iterations or trials.

```
1
2
3  ## Iteration 10; Success: False
4  def controller(obs):
5      """
6      A feedback controller that computes the action based on the observation.
7
8      Args:
9          obs: (dict) The observation from the environment.
10     Output:
11         action: (list or nd.array) A 4-dimensional vector.
12     """
13     hand_pos = obs['hand_pos']
14     puck_pos = obs['puck_pos']
15     goal_pos = obs['goal_pos']
16     gripper_open = 1 if obs['gripper_distance_apart'][0] > 0.5 else 0
17
18     normalize = lambda x: [i / max(abs(max(x, key=abs)), 1) for i in x]
19
20     # Close the gripper when close to the puck and the gripper is open
21     if gripper_open and ((abs(hand_pos[0] - puck_pos[0]) < 0.05) and (abs(hand_pos[1] -
        puck_pos[1]) < 0.05) and (abs(hand_pos[2] - puck_pos[2]) < 0.05)):
22         return [0, 0, 0, 1]
23     # When the puck is grasped, move towards the goal position with a normalized direction
24     elif not gripper_open and ((abs(hand_pos[0] - puck_pos[0]) < 0.1) and (abs(hand_pos[1] -
        puck_pos[1]) < 0.1) and (abs(hand_pos[2] - puck_pos[2]) < 0.1)):
25         direction_to_goal = [goal_pos[0] - hand_pos[0], goal_pos[1] - hand_pos[1],
        goal_pos[2] - hand_pos[2]]
26         return normalize(direction_to_goal) + [0]
27     # When the gripper is open and not close enough to the puck, move towards the puck
28     else:
29         direction_to_puck = [puck_pos[0] - hand_pos[0], puck_pos[1] - hand_pos[1],
        puck_pos[2] - hand_pos[2]]
30         return normalize(direction_to_puck) + [0]
31
32 ## Iteration 13; Success: True
33 def controller(obs):
34     """
35     A feedback controller that computes the action based on the observation.
36
37     Args:
38         obs: (dict) The observation from the environment.
39     Output:
40         action: (list or nd.array) A 4-dimensional vector.
41     """
42     hand_pos = obs['hand_pos']
43     puck_pos = obs['puck_pos']
44     goal_pos = obs['goal_pos']
45     gripper_open = 1 if obs['gripper_distance_apart'][0] > 0.5 else 0
46
47     normalize = lambda x: [i / max(abs(max(x, key=abs)), 1) for i in x]
48     # Close the gripper when close to the puck and the gripper is open
49     if gripper_open and ((abs(hand_pos[0] - puck_pos[0]) < 0.05) and (abs(hand_pos[1] -
        puck_pos[1]) < 0.05) and (abs(hand_pos[2] - puck_pos[2]) < 0.05)):
50         action = [0, 0, 0, 1]   # Close the gripper
51     elif not gripper_open and ((abs(hand_pos[0] - puck_pos[0]) < 0.1) and (abs(hand_pos[1] -
        puck_pos[1]) < 0.1) and (abs(hand_pos[2] - puck_pos[2]) < 0.1)):
52         # When the puck is grasped, adjust direction towards the goal with improved
        precision and ensure successful pickup
53         direction_to_goal = normalize([goal_pos[0] - hand_pos[0], goal_pos[1] - hand_pos[1],
        goal_pos[2] - hand_pos[2]])
54         action = direction_to_goal + [1]   # Keep the gripper closed
55     else:
56         # Move towards the puck first if not carrying it
57         direction_to_puck = normalize([puck_pos[0] - hand_pos[0], puck_pos[1] - hand_pos[1],
        puck_pos[2] - hand_pos[2]])
58         action = direction_to_puck + [0]   # Open the gripper to prepare for grasping
59
60     return action
```

**Figure A.18:** Learned Code for LLFBench Meta-World Pick-Place (Part 2). Functions with the same name are learned during different iterations or trials.

```
1  # Iteration 1
2  {
3     "reasoning": "The task involves controlling a robotic arm to perform a pick-place
       operation by interpreting observations and generating appropriate control actions (4D
       vectors) to move the arm towards the goal. The objective is to adjust the controller's
       code (__code0) based on feedback received at each step of the process. The feedback within
       #Others suggests whether the movements made by the arm are leading it closer to or
       farther from the goal, along with advice on what coordinates to aim for next (e.g.,
       [-0.07  0.68  0.12  0.  ]). The controller's code currently returns a static action [0,
       0, 0, 0], which does not adjust based on observations and hence does not efficiently
       move the robotic arm towards the goal. Given the repetitive feedback suggesting to go to
       [-0.07  0.68  0.12  0.  ], it appears that the controller needs to dynamically compute
       actions based on the difference between the current positions (hand_pos, puck_pos) and
       the goal_pos, along with possibly manipulating the gripper state for pick and place
       operation.",
4     "answer": "",
5     "suggestion": {...}
6  }
7
8
9  # Iteration 5
10 {
11    "reasoning": "The instruction involves creating a controller for a Sawyer robot arm to
       solve a pick-and-place task by providing sequential control actions based on
       observations of the environment and state of the robot. The feedback indicates that
       despite the actions taken, the success criteria have not been met, with the latest
       attempt achieving a return of 0.9574427394250022. The code provided for __code0 defines
       a controller function using the observed state to compute control actions aimed at
       grasping a puck and moving it to a goal position while adjusting the gripper based on
       its distance from the puck. Feedback throughout the execution indicates that the
       controller adapts its strategy based on proximity to the puck and whether it has been
       picked up. Feedback also suggests that the gripper's opening and closing actions are
       central to the task's success, with corrections being made to close the gripper when
       necessary. The controller's logic for determining movements toward the puck or goal and
       adjusting the gripper state appears reasonable, but the feedback implies that
       adjustments might be required to ensure task completion. This could include fine-tuning
       the conditions under which the gripper is closed or adjusting the thresholds for puck
       pickup and delivery to the goal.",
12    "answer": "",
13    "suggestion": {}
14
15 # Iteration 10
16 {
17    "reasoning": "The task involves programming a robot arm to pick up a puck and place it
       at a goal position by providing sequential 4-dimensional action vectors. The action
       vectors represent target positions and gripper states (open or close) based on the
       robot's and puck's current positions. The feedback indicates that despite optimizing the
       controller logic to close the gripper when near the puck, move towards the puck when
       it's not grasped, and head towards the goal once the puck is grasped, the actions have
       not resulted in task completion. Specifically, repeated suggestions to close the
       gripper (indicating it remained open when it should have been closed) and guidance on
       where to move next suggest that the action selection logic might not be effectively
       responding to the environment's state or might not be properly engaging the gripper at
       the right times. The latest code includes a condition to normalize the direction of
       movement and checks for gripper position, but there seems to be a misunderstanding or
       oversight in adjusting the robot's behavior based on the scenario to successfully
       complete the pick-place task.",
18    "answer": "",
19    "suggestion": {...}
20 }
21 }
```

**Figure A.19:** Example Responses of LLM for LLFBench Meta-World Pick-Place (Part 1); Content in suggestion is omitted for space (represented as ...); please see the code examples.

```
1 # Iteration 12
2 {
3     "reasoning": "In this task, the controller function for the Sawyer robot arm manipulates
        the robot's actions based on the observed environment to achieve a pick-and-place task.
        The function correctly defines actions based on the robot's hand position, the puck's
        position, the goal position, and the gripper's state. However, the feedback after each
        step in the documentation indicates that although the robot is making progress towards
        the goal, there is an issue with the gripper's action state not being switched to
        'close' (1) at the crucial moment when the robot is close enough to the puck. Most
        feedback comments suggest that the robot is moving in the right direction but fails to
        grasp the puck due to the gripper not closing. This inefficiency appears to arise from
        the conditions set for closing the gripper and possibly the normalized direction
        towards the goal after the puck has been grasped. The condition for closing the gripper
        (when close enough to the puck) seems to be met, but the action selected afterwards
        does not effectively lead to puck manipulation or goal achievement in the most
        optimized manner, as indicated by the 'Feedback' messages suggesting movements that
        imply needing adjustments in action selection.",
4     "answer": "",
5     "suggestion": {...}
6 }
7
8 # Iteration 13 (Success)
9 LLM response:
10 {
11     "reasoning": "The instruction requests improvements based on feedback for a Sawyer robot
        arm's controller function, aiming to accomplish a pick-and-place task. The controller
        function accepts observations about the robot and the environment and computes an
        action to move the robot accordingly. \n\nThe feedback provided indicates the task was
        successfully completed with a suggestion that the final command should have been
        '[0.04, 0.88, 0.24, 1]', which probably indicates the precise position and gripper
        state (closed) needed to achieve the task goal. The feedback suggests the need for
        fine-tuning actions especially towards the end of the task to ensure the robot's
        gripper is appropriately positioned and the gripper state is correct for completing the
        task. Since the feedback for the current code positively notes 'Success: True' with a
        high return value, no modifications are recommended for '__code0'. \n\nPrevious feedback
        instances with lower success metrics and recommendations for change highlight the
        iterative improvements to the controller logic, specifically related to grip control
        and movement precision. The aim was to adjust the controller's response to observation
        inputs to accurately position the gripper and handle the puck. The current code
        evidently manages these aspects well by normalizing directions and managing the
        gripper's open-close state effectively under varying circumstances (proximity to puck
        or goal and whether the puck is already grasped). These adjustments have likely led to
        the successful output in the described problem.",
12    "answer": "No changes are recommended as the feedback indicates successful task
        completion and suggests the controller's current logic adequately addresses the task's
        requirements.",
13    "suggestion": {}
14 }
```

**Figure A.20:** Example Responses of LLMs for LLFBench Meta-World Pick-Place (Part 2); Content in suggestion is omitted for space (represented as ...); please see the code examples.

