# OpenReview forum: "Trace is the Next AutoDiff: Generative Optimization with Rich Feedback, Execution Traces, and LLMs"
_NeurIPS.cc/2024/Conference — NeurIPS 2024 poster_

### Official Review · Reviewer_GPUE · 2024-06-28

**Soundness:** 4
**Presentation:** 4
**Contribution:** 4
**Rating:** 8
**Confidence:** 4

**Summary:**

The paper proposes a problem specification (OPTO) and a solution (optimizer OptoPrime) as well as a software framework (Trace) for agentic program optimization. The paper demonstrates high performance of the bundle above in solving 5 benchmarks. The paper demonstrates the ability to optimize 3 types of parameters: numeric, text prompts, and python code. The paper proposes Trace as an analogue of imperative-style AutoDiff engines like PyTorch for agentic programs relying on (black box) LLM inferences. The evaluated benchmarks cover a reinforcement learning setting (Battleship, Traffic control and Robot manipulation), scalar regression (Numerical optimization) and question answering (classification and more in BigBenchHard).

**Strengths:**

[1] The authors do a great job in demonstrating the operational performance of Trace on 5 examples: Battleship, Numerical optimization, Traffic control, BigBenchHard, and Robot manipulation.

[2] Great to see Trace compared with both traditional optimizers and the recent LLM-based (OPRO) in Figure 5b.

[3] It is nice to see that the proposed method is designed to generalize to several types of payload (demonstrated 3: numeric values, text and code).

[4] Experiments in “5.3 Unifying Prompts and Functions Optimization” neatly showcase joint optimization of prompts and code which is a clear novelty, even though one may argue that a text prompt and a text of a piece of code are both textual information. Good to see that the optimization works even without the history like Yang 2024 (OPRO) or population like Pryzant 2023 (ProTeGi).

[5] The learning seems very sample-efficient with the reported scores achieved in just several single-sample iterations.

[6] After running exp/run_prompt_bigbench.py I confirm the reproducibility of the results on BBH.

**Weaknesses:**

[1] The paper does not compare with “TEXTGRAD: Automatic Differentiation via Text” (arXiv:2406.07496v1) which is a concurrent work.

[2] The narrative goes from describing a piece of software (Trace) to the problem specification (OPTO) and then to the algorithmic solution (OptoPrime). I would expect the problem specification and the proposed algorithm to be explained first while the software details are discussed later or even in the Appendix.

[3] The experiment in 5.3 confirms that the proposed method works in principle, however due to the small scale of the optimized program (3 parameters), there is no clear signal that Trace will scale up for 10s or 100s of trainable parameters.

[4] The definition of a trace (line 126) is not given. I am familiar with the verb “to trace” with respect to a symbolic program, however it is unclear what “a trace” means. From a random web page (https://www.ituonline.com/tech-definitions/what-is-an-execution-trace/):

Definition: Execution Trace
An execution trace is a record of the sequence of operations executed by a program during its run. This trace includes function calls, variable values, and the flow of control among other details.

This definition is not mathematically clear.

[5] The work lacks the analysis of Figure A.12 where the learned prompt template does not make much sense.

[6] Running the baseline exp/run_prompt_bigbench_dspy.py fails.

[7] The paper does not demonstrate the optimization success in agentic programs with memory, i.e. when the optimized class like Predict below has internal state that changes from call to call:

```
@trace_class
class Predict(LLMCallable):
```

**Questions:**

[1] On lines 79-80 you claim that “Remarkably, there is 80 no mention of Battleship nor details on how the functions reason and act should behave or adapt 81 in Fig. 2a”. However in Appendix H, Iteration 0 (Initialization) you provide relatively detailed instructions: “Given a map, analyze the board in a game. On map, O denotes misses, X denotes successes, and . denotes unknown positions.”. Since the Battleship game is a very well known one, ChatGPT could reproduce the code memorized from the public repositories. How do you assess the risk of this type of leak?

[2] Where can an example of prompt optimization of an LLM agent as per line 767 be found?

[3] I appreciate quoting “Complete” on Figure 2a as the example is not clear without `__call__()` and `select_coordinate()` that can only be found in the supplementary materials in exp/battleship_exp.py -> Policy2. I suggest replacing it with “An excerpt from”.

[4] In Figure 3 it is not clear how can g1, g2 and g3 be different since the connectivity of the graph is defined by the optimized torch-style module, specifically a chain of 2 optimizable nodes Reason and Act in the example.

[5] It would be valuable to know the USD cost of OpenAI API usage per experiment.

[6] On the technical side, python version is not mentioned, datasets and ray are not on the requirements list. However I appreciate providing OAI_CONFIG_LIST_sample.

**Limitations:**

[1] The authors honestly admit that one of the main limitations of the current approach is LLM context length. Indeed, the scaling of the proposed algorithm that packs the entire trace into a single context for LLM inference inside OptoPrime.

[2] Scaling of the proposed framework to more learnable parameters and more sophisticated agentic programs is yet to be demonstrated.

---

> ### Author Rebuttal · Authors · 2024-08-07
>
> Thank you for your insightful comments and questions.
> # Comparison with TextGrad
> Please note that we submitted Trace to NeurIPS on May 15 2024, and TextGrad was uploaded to arXiv and Github on June 11 2024. So a direct comparison was impossible.
> # Narrative
> We considered the narrative you suggested and decided that we needed a very strong motivation for an entirely new approach to optimization (Section 1.3). We found that showing the Trace software in action first was the fastest way to demonstrate that this kind of optimization (OPTO) is feasible, practical and powerful.
> # Limitation on Scalability
> We agree that scalability is a limitation of OptoPrime, as discussed in Sec 6 and Sec 7. We wish to clarify that this limitation pertains specifically to OptoPrime and not to Trace itself. Trace is designed to scale efficiently to large graphs with minimal overhead (see line 242) and can handle non-textual nodes.
> We are inspired by the historical development of back-propagation, starting from optimizing networks with 10s of neurons (in the Rumelhart et al paper in 1986) to billions today. We anticipate better and scalable algorithms for OPTO developed in the future.
> # Definition
> We will include a clear definition of an execution trace. The execution trace is defined as the sequence of operations and their execution results invoked when computing the output from a set of inputs. This execution trace can be expressed as a computational graph defined in Preliminary. We will clarify that the DAG g is the computational graph defined above.
> # Analyzing learned prompts
> We will highlight the learned prompts in the revised Sec 5. One lesson from several automatic prompt engineering works is that human intuition is not a good guide for prompt engineering. Seemingly innocuous (and perhaps unreasonable) changes to a prompt template (like that in Figure A.12) can have large effects on an LLM’s behavior, thus motivating the need for their automatic optimization e.g. via Trace.
> # Bugs in Running DSpy baseline
> Thank you for running our code. Unfortunately, the code uploaded to OpenReview included a stale version of the DSPy script. We apologize for the oversight. We provide a snippet of our correct setup in the one-page pdf. We will update the supplementary material.
> # Optimizing stateful agents
> Optimizing stateful agents is similar to optimizing recurrent neural networks; we need to explicitly represent the state as inputs to the learned functions (discussed in line 193). Our experiments on Meta-World and Battleship are examples of stateful problems. In these problems, the environment is stateful and that state is returned as input to the functions that Trace’s learning; these graphs are similar to that of optimizing an agent with an internal state. We hope these experiments are sufficient to address the reviewer’s concern.
> # Battleship game
> We independently found this error after the paper submission. The exact function definition used in the experiments is given in Appendix H, which is consistent with the code we submitted. You are right that Figure 2a over-simplifies important details about the function docstring and the current text in line 79 is misleading. We will change it to “Remarkably, there is no mention of Battleship environment APIs nor details on how the functions reason and act should behave or adapt in Fig. 2a”.
>
> We agree that GPT4 likely has trained on code about the battleship game. But it does not know the API of the Battleship game (because we coded it up from scratch). If GPT4 did know how to solve the problem we presented, it could have solved it in the first iteration after one update, but that is not what we observed (Figure 1). This gap between the performance after one update and multiple updates indicate the sign and need of learning from feedback and interactions.
> We will use “excerpt” in the revision to clarify the code snippets in Figure 2.
> # LLM Prompt Optimization Example
> An example of LLM agent as per line 767 can be found in a new experiment that we conducted for the [virtualhome](http://virtual-home.org/) environment. This experiment is included in the one-page PDF for the rebuttal, including the code and figure. Virtualhome is a collaborative environment that requires two agents to work together to solve household tasks. Trace is asked to optimize and update a specific part of the prompt, which is the plan for future actions. Prior work (Guo et al., "Embodied LLM Agents Learn to Cooperate in Organized Teams", 2024) forces agents to have a round of conversation before they start the task. We show that Trace allows agents to have naturally emerging pro-social behaviors for some tasks (such as “putting plates into the dishwasher”), but not others (such as “reading a book”).
> # Graphs in Fig 3
> Figure 3 is an illustration of a general OPTO problem setup. In the Battleship example, you are correct that the graph is the same in every iteration. But, for a general optimization problem, the graph structure can be different, e.g. different parameters changing program flow during the forward pass, or simply because the execution is stochastic. Meta-World is an example where the graph structure can be different across iterations. The graph is a chain describing the multi-step interactions with the environment, and each episode ends either when the robot successfully solves the problem or when timeout happens. Therefore, iterations with successful episodes can have a shorter chain than those that fail and time out.
> # Token Cost
> The cost depends on the graph size and the tokens required to describe the problem. Running OptoPrime with GPT-4-Turbo for the most complex experiment in the paper, MetaWorld, costs <$30 USD for one task (over 10 seeds, 30 iterations). Costs of other experiments are a fraction of this.
> # Technical Details
> We currently require Python>=3.8, and we will update the setup.py to add the missing dependencies. Thank you for your feedback and running the code!

---

> > ### Comment · Reviewer_GPUE · 2024-08-12
> >
> > Thank you for addressing the weaknesses and questions. I intend to keep my score.

---

### Official Review · Reviewer_aQXt · 2024-07-12

**Soundness:** 2
**Presentation:** 2
**Contribution:** 2
**Rating:** 5
**Confidence:** 3

**Summary:**

This paper proposes an end-to-end optimization framework, Trace, for the automatic design and updating of artificial intelligence systems. Trace is based on Optimization with Trace Oracle (OPTO), treating the computational workflow of AI systems as a graph of neural networks, which can be updated via backpropagation. Additionally, this paper introduces OptoPrime, a general optimizer based on large language models (LLM), as a specific implementation of Trace. In the experimental section, the paper compares Trace with the state-of-the-art LLM optimizer OPRO across various tasks, including numerical optimization, traffic control, and robotic control. The results demonstrate the superior performance of Trace. The primary contribution of this work lies in modeling computational workflows as OPTO problems and designing Trace + OptoPrime to address these issues.

**Strengths:**

Strengths:
1.Clarity of Writing: The structure of the article is clear, the language is fluent, and it is easy to understand.
2.Novelty: This paper introduces a novel end-to-end optimization framework, Trace, which views the computational process as a graph and utilizes execution trace information to optimize parameters. Compared to traditional black-box optimization methods, this approach offers higher efficiency and greater interpretability.

**Weaknesses:**

Weaknesses:
1.Limited Scalability: The OptoPrime optimizer mentioned in the paper has some scalability limitations, such as difficulty in handling parameters or nodes that cannot be represented textually, and challenges in dealing with computational workflows that contain a large number of nodes.
2.The graph for Trace requires manual design, lacking automated methods.
3.Limited Experimental Improvement: The experimental section shows limited improvement, with the differences between Trace and OPRO not being particularly significant.

**Questions:**

1.Difference Between Graph Optimization and Individual Node Optimization: Graph optimization considers the entire computational workflow as an interconnected system, optimizing parameters in a holistic manner, which can lead to more coordinated and efficient results. In contrast, optimizing individual nodes treats each component in isolation, potentially missing out on interactions between nodes. However, the paper lacks experimental evidence to support the effectiveness of Trace in performing graph optimization over individual node optimization.
2.Discrepancy Between Ablation Studies in Figure 5 and Figure 6: The conclusions drawn from the ablation studies in Figure 5 and Figure 6 differ, raising questions about consistency.
3.Length of Trace's Prompts Compared to OPRO and Token Efficiency: The prompts used by Trace are longer than those used by OPRO. It is important to quantify this difference in length to assess its impact on token efficiency.

**Limitations:**

Yes

---

> ### Author Rebuttal · Authors · 2024-08-07
>
> Thank you for your insightful comments and questions.
>
>
> # Limited Scalability
>
> We agree that scalability is a limitation of OptoPrime, as discussed in Section 6 (Limitations) and Section 7 (Conclusion) where we note the current focus on textualizable problems. However, we wish to clarify that this limitation pertains specifically to OptoPrime and not to Trace itself. Trace is designed to scale to large graphs with minimal overhead (see line 242) and can handle non-textual nodes.
> OptoPrime's limitation arises from converting the propagated Trace graph into a single query that fits in the context window for a current-gen LLM. As its name suggests, OptoPrime is a first step in solving OPTO problems. We are inspired by the historical development of back-propagation, starting from optimizing networks with 10s of neurons (in the Rumelhart et al paper in 1986) to billions today. We anticipate better and scalable algorithms for OPTO developed in the future, such as using multi-agent workflows to scale to large graphs and VLMs to interpret multi-modal parameters and feedback.
>
> # Lacking an automated method
>
> We wish to clarify that Trace constructs the computational graph automatically (line 183) while the program executes (dynamically like in PyTorch), rather than requiring users to pre-define it or pre-compile it (statically like in Tensorflow 1). This means that the created graph changes automatically based on the workflow process, with different inputs potentially resulting in different graph structures.
> In Section 3.1, we discuss how to abstract the workflow using @bundle operators. This abstraction is different from manually defining the graph, which may be the source of the confusion. The @bundle decorator is an optional feature that allows users to simplify the workflow’s text representation (e.g. to reduce input token costs), enabling LLMs to better understand and optimize the workflow.
>
> # Limited Experimental Improvement
>
> We respectfully disagree on the comment on the performance gap between OptoPrime and OPRO. In almost all experiments (Figure 1, 5b, 6a, 6b), OptoPrime is a significant improvement over OPRO, e.g. 2x-4x improvement in success or rewards. Only the experiment in Figure 6c shows the different algorithms’ performance is within the error margin.
>
> # Difference between Node and Graph Optimization
>
> Consider the following example that shows why optimizing over a graph is better than optimizing only individual nodes.
> ```python
> @bundle()
> def function1(x):
>     return x > 0
> @bundle()
> def function2(y):
>     return y % 2 == 0
> def xor_test(x, y):
>     return function1(x).neq(function2(y))
> input1 = node(3, trainable=True); input2 = node(4, trainable=True)
> xor_test(input1, input2).backward(feedback=”Find a set of inputs to make the return True.”)
> ```
> When we optimize an individual node, we only see that node input, the function output, and the feedback. We do not see the other inputs and how they can affect the outcome. When we optimize for the full graph however, we can see that each input only partially affects the outcome, and we need to jointly optimize both inputs to achieve a desired outcome.
>
> # Discrepancy between Fig 5 and Fig 6’s Conclusions
> In all the experiments, OptoPrime with memory (denoted as Trace) performs better than OptoPrime without memory (denoted as Trace NoMem) and OptoPrime with memory but with the execution trace info removed from the prompt (denoted as Trace Masked). We noticed that the current writing is not clear about what each method means, which may cause confusion. Across the different ablations and Figure 5 and 6, we consistently see that memory improves performance, and masking the execution trace information hurts performance. We will better clarify the ablations in the revision.
>
> # Token Efficiency
> Thank you for the excellent suggestion, we will include token counts for the OPRO and OptoPrime prompts in the paper. Here are the statistics for the prompts at the first iteration of optimization (note OPRO's token usages grows with iterations):
>
> | Domain    | OPRO | OptoPrime |
> | -------- | ------- | ------- |
> | Numerical Opt | 175 | 918 |
> | BigBench-Hard | NA | 1883 |
> | Traffic Opt | 198 | 1679 |
> | MetaWorld | 470 | 7101 |
> | Battleship | 437  |  1305  |
>
> We can see that indeed OptoPrime consumes significantly more tokens than OPRO. However, we observe consistently that even allowing 7-10x more iterations of OPRO so as to equalize token costs, the OPRO performance plateaus to a worse level than OPTOPrime (e.g. Figure 1: OPRO at Iter 7 vs. OptoPrime at Iter 2; Figure 5b: OPRO at Iter 50 vs. OptoPrime at Iter 5; Figure 6b: OPRO at Iter 30 vs. OptoPrime at Iter 10, etc.). OPRO is suboptimal not due to a token limit but instead a lack of information, which is captured and represented using Trace.

---

### Official Review · Reviewer_YuBN · 2024-07-13

**Soundness:** 4
**Presentation:** 4
**Contribution:** 3
**Rating:** 7
**Confidence:** 2

**Summary:**

The paper introduces Trace, a novel optimization framework that instances the concept of Optimization with Trace Oracle (OPTO). In Trace, the computational workflows is treated as dynamic graphs and rich information, including intermediate results, processing details and computational graph, are used as feedback for optimization instead of traditional gradients. The framework includes a general-purpose optimizer, OptPrime, to solve the OPTO problem. In the experiment section, Trace is shown to be comparable to the first-order gradient optimizer on a numerical optimization task, and outperform baseline LLM-based methods across a wide range of tasks including , hyper-parameter tuning, robot controller design, etc. Additionally , Trace offers a Python interface that can integrate seamlessly with PyTorch.

**Strengths:**

1.I think using execution traces instead of gradients for optimization is quite innovative. This allows for the optimization of workflows that are non-differentiable or have dynamic computation graphs.

2.This framework can be applied to a wide range of tasks, including robotics and AI systems.

3.Trace has shown superior performance compared to other LLM-based optimizers and has demonstrated results comparable to those of traditional gradient-based optimization methods.

4.The Python interface for Trace simplifies integration with existing codebases.

**Weaknesses:**

As someone outside of this field, I find this paper to be quite impressive.  I did not identify any specific weaknesses.

**Questions:**

1. Could Trace be adapted for use with other programming languages, and what would such an adaptation entail in terms of architectural changes?

**Limitations:**

Yes

---

> ### Author Rebuttal · Authors · 2024-08-07
>
> Thank you for your insightful comments and question.
>
> Yes, in principle, Trace can indeed be adapted for use with other programming languages. The core design of Trace is based on the primitives node and @bundle, which define the nodes and operators, respectively, for the directed acyclic graph (DAG) abstraction of the traced computational process. Once the DAG is created in any programming environment, the algorithms used in Trace can be applied.
>
> In our current implementation, we overloaded Python’s magic methods to seamlessly integrate with existing Python code. This approach may not be feasible in other programming languages due to the limitations of operator overloading, which could result in a less clean interface. Nonetheless, by building a set of operators using the idea of @bundle, we can create DAGs to abstract the computational process in other languages. One example demonstrating the feasibility of such an adaptation is the C++ versions of AutoDiff libraries like PyTorch (which are also based on DAGs).
>
> Therefore, we believe that the DAG-based design that Trace employs can be effectively adapted to other programming languages. However, we acknowledge that developing Trace libraries for other languages can require non-trivial engineering. We hope that the impressive results demonstrated in our paper will inspire future development to adapt Trace to a broader range of programming environments.

---

> > ### Comment · Reviewer_YuBN · 2024-08-11
> >
> > Thank you for your detailed response and congratulations on your excellent work!!

---

### Author Rebuttal · Authors · 2024-08-07

Thank you for reviewing this paper. This PDF contains figures and codes of the new virtual home experiments and a correction on the submitted code for running DSpy baseline, which is for addressing Reviewer GPUE's questions.

Virtualhome is a collaborative, stateful environment that requires two LLM agents to work together to solve household tasks. Trace is asked to optimize and update a specific part of the prompt, which is the plan for future actions. Prior work (Guo et al., "Embodied LLM Agents Learn to Cooperate in Organized Teams", 2024) forces agents to have a round of conversation before they start the task. We show that Trace allows agents to have naturally emerging pro-social behaviors for some tasks (such as “putting plates into the dishwasher”), but not others (such as “reading a book”).

---

### Decision · Program_Chairs · 2024-09-25

**Decision:**

Accept (poster)

**Comment:**

This paper introduces a novel end-to-end optimization framework, Trace, which views the computational process as a graph and utilizes execution trace information to optimize parameters. As a case study, this paper also developed a general-purpose optimizer, OptoPrime that can effectively solve OPTO problems,such as  first-order numerical optimization, prompt optimization, hyper-parameter tuning, robot controller design, code debugging, etc.,

The limitations of the works lie in the scalability such as the context length and graph size.  Despite some concerns about scalability and the need for clearer experimental results, the strengths of the paper—especially its novelty, versatility, and demonstrated performance—outweigh these weaknesses. Therefore, I recommend acceptance, with the suggestion that the authors consider revising the narrative structure and enhancing the clarity of experimental results in the final version.